

# Atmospheric histories and emissions of chlorofluorocarbons CFC-13 (CClF$_3$), CFC-114 (C$_2$Cl$_2$F$_4$), and CFC-115 (C$_2$ClF$_5$)

 Martin K. Vollmer[1],  Dickon Young[2],  Cathy M. Trudinger[3],  Jens Mühle[4],  Stephan Henne[1],  Matthew Rigby[2],  Sunyoung Park[5],  Shanlan Li[5],  Myriam Guillevic[6],  Blagoj Mitrevski[3],  Christina M. Harth[4], Benjamin R. Miller[7,8],  Stefan Reimann[1],  Bo Yao[9],  L. Paul Steele[3],  Simon A. Wyss[1],  Chris R. Lunder[10],  Jgor Arduini[11,12],  Archie McCulloch[2],  Songhao Wu[5],  Tae Siek Rhee[13],  Ray H. J. Wang[14], Peter K. Salameh[4],  Ove Hermansen[10],  Matthias Hill[1],  Ray L. Langenfelds[3],  Diane Ivy[15],  Simon O'Doherty[2],  Paul B. Krummel[3],  Michela Maione[11,12],  David M. Etheridge[3],  Lingxi Zhou[16],  Paul J. Fraser[3],  Ronald G. Prinn[15],  Ray F. Weiss[4], and  Peter G. Simmonds[2]

[1]Laboratory for Air Pollution and Environmental Technology, Empa, Swiss Federal Laboratories for Materials Science and Technology, Überlandstrasse 129, 8600 Dübendorf, Switzerland
[2]Atmospheric Chemistry Research Group, School of Chemistry, University of Bristol, Bristol, UK.
[3]Climate Science Centre, CSIRO Oceans and Atmosphere, Aspendale, Victoria, Australia.
[4]Scripps Institution of Oceanography, University of California at San Diego, La Jolla, California, USA.
[5]Kyungpook Institute of Oceanography, Kyungpook National University, South Korea.
[6]METAS, Federal Institute of Metrology, Lindenweg 50, Bern-Wabern, Switzerland.
[7]Earth System Research Laboratory, NOAA, Boulder, Colorado, USA.
[8]Cooperative Institute for Research in Environmental Sciences, University of Colorado, Boulder, Colorado, USA.
[9]Meteorological Observation Centre (MOC), China Meteorological Administration (CMA), Beijing, China.
[10]Norwegian Institute for Air Research, Kjeller, Norway.
[11]Department of Pure and Applied Sciences, University of Urbino, Urbino, Italy.
[12]Institute of Atmospheric Sciences and Climate, Italian National Research Council, Bologna, Italy.
[13]Korea Polar Research Institute, KIOST, Incheon, South Korea.
[14]School of Earth and Atmospheric Sciences, Georgia Institute of Technology, Atlanta, Georgia, USA.
[15]Center for Global Change Science, Massachusetts Institute of Technology, Cambridge, Massachusetts, USA.
[16]Chinese Academy of Meteorological Sciences (CAMS), China Meteorological Administration (CMA), Beijing, China.

*Correspondence to:* Martin K. Vollmer (martin.vollmer@empa.ch)

**Abstract.** Based on observations of three chlorofluorocarbons, CFC-13 (chlorotrifluoromethane), CFC-114 (dichlorotetrafluoroethane) and CFC-115 (chloropentafluoroethane) in atmospheric and firn samples, we reconstruct records of their tropospheric histories spanning nearly eight decades. These compounds were measured in polar firn air samples, in ambient air archived in canisters, and in-situ at the AGAGE (Advanced Global Atmospheric Gases Experiment) network and affiliated sites. Global

5 emissions to the atmosphere are derived from these observations using an inversion based on a 12-box atmospheric transport model. For CFC-13, we provide the first comprehensive global analysis. This compound increased monotonically from its first appearance in the atmosphere in the late 1950s to a mean global abundance of 3.18 ppt (dry air mole fraction in parts-per-trillion, pmol mol$^{-1}$) in 2016. Its growth rate has decreased since the mid 1980s but has remained at a surprisingly high level of 0.02 ppt yr$^{-1}$ since the late 2000s. CFC-114 increased from its appearance in the 1950s to a maximum of 16.6 ppt in the

10 early 2000s, and has since slightly declined to 16.3 ppt in 2016. CFC-115 increased monotonically from its first appearance





in the 1960s and reached a global mean mole fraction of 8.52 ppt in 2016. Growth rates of all three compounds over the past years are significantly larger than would be expected from zero emissions. Under the assumption of unaltered lifetimes and atmospheric transport patterns, we derive global emissions from our measurements, which have remained unexpectedly high in recent years: Mean yearly emissions for the last decade (2007–2016) of CFC-13 are at 0.48 ±0.15 kt yr$^{-1}$ (>15% of past

peak emissions), of CFC-114 at 1.90 ±0.84 kt yr$^{-1}$ (∼10% of peak emissions), and of CFC-115 at 0.80 ±0.50 kt yr$^{-1}$ (>5% of peak emissions). Mean yearly emissions of CFC-115 for 2014–2016 are 1.08 ±0.50 kt yr$^{-1}$ and have more than doubled compared to 2009. Cumulative global emissions for CFC-114 derived from observations through 2016 exceed the global cumulative production derived from reported inventory data by >10% while those for CFC-115 agree well. We find CFC-13 emissions from aluminum smelters and impurities of CFC-115 in the refrigerant HFC-125 (CHF$_2$CF$_3$) but if extrapolated to

global emissions neither of them can account for the lingering global emissions determined from the atmospheric observations. We also conduct regional inversions for the years 2012–2016 for the north-east Asian area using observations from the Korean Gosan AGAGE site and find significant emissions for CFC-114 and CFC-115, suggesting that a large fraction of their global emissions currently occur in north-eastern Asia and more specifically on the Chinese mainland.

## 1    Introduction

Chlorofluorocarbons (CFCs) are very stable man-made compounds known to destroy stratospheric ozone. For this reason they were regulated for phase-out under the Montreal Protocol on Substances that Deplete the Ozone Layer, and its subsequent amendments. The ban has been effective since the end of 1995 for developed countries and the end of 2010 for developing countries. The ban is put on production for emissive use and does not cover production for feedstock, or recycling of used CFCs for recharging of old equipment, the latter being applied particularly in the refrigeration sector. While

the dominant CFCs in the atmosphere are CFC-12 (CCl$_2$F$_2$), CFC-11 (CCl$_3$F), and CFC-113 (C$_2$Cl$_3$F$_3$), this article reports on CFC-13 (chlorotrifluoromethane, CClF$_3$), CFC-114, here defined as the combined isomers 1,2-dichlorotetrafluoroethane (CClF$_2$CClF$_2$, CFC-114, CAS 76-14-2) and 1,1-dichlorotetrafluoroethane (CCl$_2$FCF$_3$, CFC-114a, CAS 374-07-2), and CFC-115 (chloropentafluoroethane, C$_2$ClF$_5$). The compounds were mainly used in specialized refrigeration, hence their abundances in the atmosphere are considerably smaller than those of the three major CFCs. However, their atmospheric lifetimes are sig-

nificantly longer (see Table 1 for climate metrics). Ozone depletion potentials (ODPs) for the three compounds are high and their radiative efficiencies are high, thereby yielding high global warming potentials (GWPs), with that for CFC-13 (13 900, GWP-100yr) only surpassed by very few other greenhouse gases.

Removal of these CFCs from the atmosphere occurs predominantly in the stratosphere through ultraviolet (UV) photolysis and reaction with excited atomic oxygen (O($^1$D)), and to a lesser extent by Lyman-$\alpha$ photolysis in the mesosphere. The at-

mospheric lifetime for CFC-13 of 640 yr used in the present study is based on a study by Ravishankara et al. (1993) and is dominated (80%) by the removal through reaction with O($^1$D), see Table 1. The lifetimes for CFC-114 and CFC-115 have recently been revised as part of the SPARC (Stratosphere-troposphere Processes And their Role in Climate) lifetimes assessment (SPARC, 2013). In that study the lifetime of CFC-114 has been reported as 189 yr (153–247 yr) with 72% of the loss from





UV-photolysis and 28% from reaction with O($^1$D) (Burkholder and Mellouki, 2013). For CFC-115 the lifetime has been significantly reduced from 1700 yr in earlier studies (Ravishankara et al., 1993) to 540 yr (404–813 yr) (SPARC, 2013) mainly due to significantly revised O($^1$D) kinetics (Baasandorj et al., 2013). This revised lifetime gives 37% of the loss derived from UV-photolysis and 63% from reaction with O($^1$D) and a minor contribution from Lyman-$\alpha$ photolysis (Burkholder and Mellouki,
2013).

CFC-13 and its R-503 blend with 40% by mass of HFC-23 (CHF$_3$) have been used as special-application low-temperature refrigerants (Calm and Hourahan, 2011; IPCC/TEAP, 2005) but small enhancements in CFC-13 were also found in the emissions from aluminum plants (Penkett et al., 1981; Harnisch, 1997). CFC-13 could also be present as an impurity in CFC-12 (CCl$_2$F$_2$) due to over-fluorination during production.

Reports on atmospheric CFC-13 in peer-reviewed articles are rare. Early measurements were reported on by Rasmussen and Khalil (1980), Penkett et al. (1981) and Fabian et al. (1981), who measured a first atmospheric vertical profile of this compound. CFC-13 measurements were made by Oram (1999) in the samples of the Southern Hemisphere Cape Grim Air Archive (CGAA) covering 1978 – 1995. He found increasing mole fractions from 1.2 pptv (therein reported in parts-per-trillion by volume) in 1978 to 3.5 pptv in 1995. Emissions deduced for this period peaked in 1987 at 3.6 kt yr$^{-1}$. Culbertson et al. (2004) published
long records of CFC-13 measurements in background air from stations in the USA and Antarctica. For their longest record from Cape Meares (Oregon, USA), they reported on a near-constant growth of CFC-13 for the earlier part of the record with the tropospheric abundance leveling off in the late 1990s at ~3.5 pptv.

CFC-114 was used as refrigerant, blowing agent, and aerosol propellant (Fisher and Midgley, 1993; IPCC/TEAP, 2005). CFC-114 is listed as a refrigerant in blends R-400 with CFC-12 in various proportions, and in R-506 with 55% by mass
HCFC-31 (CH$_2$ClF) (Calm and Hourahan, 2011; IPCC/TEAP, 2005). It was also used unblended in specialized refrigeration e.g. in U.S. naval equipment from where it was phased out over the course of several decades following the Montreal Protocol ban (Toms et al., 2004; IPCC/TEAP, 2005). Uranium isotope effusion is a process that, at least in the past, involved significant amounts of CFC-114 for cooling, but now PFCs are used as a substitute (IPCC/TEAP, 2005).

Some of the first CFC-114 measurements were conducted at urban sites in the 1970s by Singh et al. (1977) who reported ele-
vated mole fraction up to 170 ppt (parts-per-trillion or picomol mol$^{-1}$). Measurements in background air followed (Singh et al., 1979) and a transect across the equator in 1981 showed a global mole fraction of 14 ppt (Singh et al., 1983). In the early 1980s Fabian et al. (1981, 1985) measured vertical profiles of CFC-114 in the atmosphere and found a decreasing mole fraction from 10.5 pptv at 10 km to 2.7 pptv at 35 km. Hov et al. (1984) measured CFC-114 of 10.9 ppt in samples collected from Spitsbergen in spring 1983. Schauffler et al. (1993) reported on measurements of CFC-114 near the tropical tropopause and also
Chen et al. (1994) measured vertical profiles and a first multi-year record in both hemispheres showing increases of CFC-114 at Hokkaido from 10 pptv in 1986 to 15 pptv in 1993, and a first indication of a slow-down of the atmospheric growth. This was also the first group that separated CFC-114 from CFC-114a. Oram (1999) also separated the two isomers and measured records from the CGAA covering 1978 – 1995 showing increases from 8.5 pptv to 16.5 pptv for CFC-114 and from 0.55 pptv to 1.75 pptv for CFC-114a. These results showed for the first time an increasing ratio of CFC-114a/CFC-114 in the atmosphere,
pointing to a variable ratio of their emissions over time. The first high-frequency measurements of CFC-114 from Cape Grim


for 1998 and 1999 showed an abundance of 16.7 ppt, no pollution events in the footprint of the station and no detectable trend (Sturrock et al., 2001). A first atmospheric long-term record of CFC-114 was published by Sturrock et al. (2002) based on firn air measurements from Antarctica and using the CGAA record from Oram (1999), revealing an onset of growth of this compound in the atmosphere in the early 1960s. Martinerie et al. (2009) modeled atmospheric CFC-114 records based on several firn air profiles and found a much earlier atmospheric appearance and larger abundances up to approximately 1980 compared to Sturrock et al. (2002). In a recent study, Laube et al. (2016) reconstructed atmospheric CFC-114 and CFC-114a histories of abundances and emissions based on CGAA and firn air measurements. The study confirmed the temporally variable ratio of the two isomer abundances, which translate into a CFC-114a/CFC-114 emission ratio that increased sharply in the early 1990s but gradually declined since then.

CFC-115 was used as refrigerant R-115 and occurred also in blends R-502 with 49% by mass HCFC-22 ($CHClF_2$) and R-504 with 48% by mass HFC-32 ($CH_2F_2$) (Calm and Hourahan, 2011; IPCC/TEAP, 2005; Fisher and Midgley, 1993). It has also been used as an aerosol propellant, and to a minor extent as a dielectric fluid (Fisher and Midgley, 1993). First measurements of CFC-115 were made by Penkett et al. (1981) and Fabian et al. (1981) who reported on an atmospheric vertical profile. These were later complemented by more vertical atmospheric profiles (Pollock et al., 1992; Schauffler et al., 1993; Fabian et al., 1996). Later temporal records of ground-based measurements based on flask samples were published by Oram (1999) for the CGAA and Culbertson et al. (2004) for both hemispheres. Sturrock et al. (2001) reported on the first in-situ measurements of CFC-115 at ∼8 ppt for Cape Grim for 1998/1999 with a small growth of ∼5% yr$^{-1}$. The above-mentioned firn air analysis by Sturrock et al. (2002) produced a first long-term record of CFC-115, and showed significantly higher abundances for the 1980s compared with the CGAA record measured by Oram (1999). In contrast, CFC-115 reconstruction by Martinerie et al. (2009) were much in agreement with the early results from the CGAA record (Oram, 1999).

Here we report on measurements of CFC-13, CFC-114 (combined $C_2Cl_2F_4$ isomers), and CFC-115 from the Advanced Global Atmospheric Gases Experiment (AGAGE) and affiliated networks, and from measurements in archived air samples of the CGAA and the Northern Hemisphere. We further report on measurements from air samples collected from polar firn in both hemispheres, which we interpret using a firn air model. For each of the three compounds, all measurements are made and reported against a single primary calibration scale. Our observations are used with the AGAGE 12-box model and two inversion systems to derive global emissions. We further apply an inversion system to estimate regional emissions of CFC-115 from north-eastern Asia for the years 2012 – 2016. For CFC-13, this is the first comprehensive study available on atmospheric abundances and emissions.

## 2 Methods

### 2.1 Stations and Data Records for in Situ and Flask Measurements

The present study includes in situ measurements at the stations of the AGAGE (Advanced Global Atmospheric Gases Experiment, URL: https://www.agage.mit.edu) and its affiliated networks (Fig. 1). Measurements reported here are mostly based on Medusa gas chromatography mass spectrometry (GCMS) techniques (Miller et al., 2008). In Europe, measurements are made



at Zeppelin (Ny Ålesund, Spitsbergen), Mace Head (Ireland), Jungfraujoch (Switzerland), and Monte Cimone (Italy), the latter being equipped with different instrumentation (Maione et al., 2013). Measurements are further conducted at Trinidad Head (California, USA), Ragged Point (Barbados), Cape Matatula (American Samoa) and Cape Grim (Tasmania, Australia). The East Asian region is covered by stations at Gosan (Jeju Island, South Korea) and Shangdianzi (China). In addition to these

in-situ measurements, we also include measurements of samples collected weekly since 2007 at the South Korean Antarctic station King Sejong, King George Island (South Shetland Islands) and analyzed at the Swiss Federal Laboratories for Materials Science and Technology (Empa) using Medusa-GCMS technologies (Vollmer et al., 2011). We also provide a qualitative description of measurements in urban areas from Tacolneston (Great Britain, 100 km northeast of London), Dübendorf (outskirts of Zurich, Switzerland), La Jolla (outskirts of San Diego, USA), and Aspendale (outskirts of Melbourne, Australia). At

a few AGAGE stations, measurements of CFC-114 and CFC-115 were previously made with different GCMS instrumentation (Adsorption-Desorption-System, Simmonds et al. (1995)), however the precisions and standards propagations of these early measurements are significantly poorer than those using Medusa-GCMS technology and hence these results are not included in the present analysis.

Most of the AGAGE network observations for the three CFCs are published here for the first time in a journal article. How-

ever some of the measurements have been previously used in Ozone Assessment Reports (e.g. Carpenter and Reimann, 2014), in modeling studies to derive global emissions (Rigby et al., 2014), and for Cape Grim, were reported in Baseline series starting 1997–1998 issue. For CFC-114 and CFC-115 the data are available directly from the AGAGE website (https://agage.mit.edu/) and data repositories mentioned therein.

## 2.2   Archived Air

Our analysis includes the results from Medusa-GCMS measurements of Cape Grim Air Archive (CGAA) samples collected for archival purposes since 1978 at the Cape Grim Baseline Air Pollution Station (Fig. 1). The CGAA includes >100 samples mostly collected into 34 L internally electropolished stainless steel canisters using cryogenic sampling techniques (Fraser et al., 1991; Langenfelds et al., 1996, 2014; Fraser et al., 2016). Most samples were analyzed on the Medusa-9 instrument in 2006 at CSIRO (Aspendale, Australia) using Medusa-GCMS technology with a Medusa-standard PoraBOND Q chromatography

column (Miller et al., 2008). In 2011 many samples were re-analyzed and newly-added samples were analyzed for CFC-13 and CFC-114 on the same instrument but fitted with a GasPro chromatography column (Ivy et al., 2012). In 2016 all three compounds discussed here were reanalyzed and newly-added samples were analyzed on the same instrument fitted with a GasPro column and an additional GasPro pre-column (Vollmer et al., 2016). All samples collected since 2004 are also analyzed on the Cape Grim based Medusa-3 instrument. A comparison of the different analysis sets is provided in the Supplement and

shows good agreement indicating stability of the three CFCs in the internally electropolished canisters. For the present analysis we use the mean of the measured mole fractions from these three analysis sets.

Archived air samples from the Northern Hemisphere (NH) are also included in this study. These >100 samples were collected at various sites and cover the period 1973 to present. The majority of the samples were provided by Scripps Institution of Oceanography (SIO) and collected at La Jolla and at Trinidad Head (California, USA). All samples were analyzed at SIO on



Medusa-1. These NH archive air samples were not exclusively collected for archival purposes, and potentially includes samples collected during non-background conditions (influenced by emission sources) or with non-conservative sampling techniques. Consequently a rigorous data processing was necessary to limit the record to results deemed representative of broad atmospheric regions far from emission sources (hereafter termed "background"). In particular, the earlier record of CFC-114 proved

not useful for the present analysis because there were too many anomalous sample measurement results. Numerical results for the NH and the CGAA measurements are given in the Supplement.

## 2.3  Air entrapped in Firn

Our data sets are complemented by measurements of the three CFCs in air entrapped in firn from samples collected in Antarctica and Greenland (Fig. 1). The Antarctic samples were collected in 1997–1998 at the DSSW20K site (66.77°S, 112.35°E,

1200 masl, ∼20 km west of the deep DSS drill site near the summit of Law Dome, East Antarctica (Trudinger et al., 2002; Sturrock et al., 2002)), and one deep sample originates from the South Pole in 2001 (Butler et al., 2001). The Greenland firn air samples used in the present analysis were collected near the northwest Greenland ice drill site NEEM (North Greenland Eemian Ice Drilling) at 77.45°N, 51.06°W, 2484 m.a.s.l.) in 2008 (NEEM-2008, EU hole, Buizert et al. (2012)). Due to the remote locations of these sites, these samples are considered as representative of background air. More details on these samples and on

their analysis are described by Vollmer et al. (2016) and Trudinger et al. (2016). Results for CFC-114 and CFC-115 from the DSSW20K firn air profile based on older measurement technologies and interpreted with an old version of the CSIRO firn diffusion model were previously reported by Sturrock et al. (2002) and are compared to our measurements in the Supplement.

## 2.4  Measurement Techniques and Instrument Calibration

Almost all measurements reported here are conducted with Medusa-GCMS instruments (Miller et al., 2008). Typically a sam-

ple is preconcentrated on a first cold trap filled with HaySep D and held at ∼−160°C before it is cryofocussed onto a second trap at similar temperature and in this process, remnants of oxygen and nitrogen and significant fractions of carbon dioxide and some noble gases are removed. The sample is then injected onto the chromatographic column (CP-PoraBOND Q, 0.32 mm ID × 25 m, 5 $\mu$m, Varian Chrompack, batch-made for AGAGE applications) of the GC (Agilent 6890), purged with helium (grade 6.0), which is further purified using a getter (HP2, VICI, USA). The sample is then detected in the quadrupole mass

spectrometer in selected ion mode (initially Agilent model 5973 with upgrades to model 5975 over time for most stations).

In the Medusa-GCMS, technology separation of the CFC-114 ($CClF_2CClF_2$) isomer from CFC-114a ($CCl_2FCF_3$) is not possible hence the measurements include the cumulative abundances of the two compounds. Throughout this paper we use the term "CFC-114" for the combined measurement of the two isomers. Our inability to separate the two isomers can potentially lead to biases compared to the numeric sum of their individual measurements due to a combination of two facts; one being that

the ratio of the two isomers is likely to vary both in the measured samples (Laube et al., 2016) and in the reference material used to propagate the primary calibration scales and the other being that the molar responses of the mass spectrometer are potentially different for the two isomers. We estimate a maximum potential bias, which increases from ∼0.3% for our modern record (2004–present) to ∼3.1% for the oldest samples in our archived air records (see Supplement).





For each of the three CFCs at least two fragments are routinely measured. While a target ion is used for the quantification of the peak size, the qualifying ions are mainly used for quality control by assessing the peak size ratio to the target ion, most importantly to check for potential coelution with compounds which share the target ion. CFC-13 is measured with the target ion $C^{35}ClF_2^+$ (with a mass/charge, $m/z$, 85) and the qualifying ions $CF_3^+$ ($m/z$ 69) and $C^{37}ClF_2^+$ ($m/z$ 87). On the PoraBOND Q column this compound elutes near HFC-32 and precedes ethane by ~2 sec. CFC-114 is measured with the target ion $CF_2C^{35}ClF_2^+$ ($m/z$ 135), and the target ions $CF_2C^{37}ClF_2^+$ ($m/z$ 137) and $C^{35}ClF_2^+$ ($m/z$ 85). It elutes a few seconds after H-1211 ($CBrClF_2$) and co-elutes with n-butane. CFC-115 is measured with the target ion $CF_2CF_3^+$ ($m/z$ 119) and the qualifying ions $CF_2C^{35}ClF_2^+$ ($m/z$ 135) and $CF_2C^{37}ClF_2^+$ ($m/z$ 137). It elutes ~12 sec after HFC-125 ($CHF_2CF_3$) and ~15 sec before HFC-134a ($CH_2FCF_3$). Some of the instruments are set to only acquire two fragments instead of three and for some, the sequences of target and qualifying ions are different from the above-mentioned orders.

CFC-114 measurements in strongly polluted air samples at some stations (mainly urban) have shown an analytical interference, which is believed to suppress the MS response to the quantities present in the sample. Although not fully understood, the interference is suspected to derive from large amounts of n-butane, which co-elutes with CFC-114. A decrease of 0.20 ppt is estimated for CFC-114 for an increase of 1.0 ppb (parts-per-billion, nmol mol$^{-1}$) of n-butane. The measurements of CFC-114 used in the present analysis derive from air samples not significantly polluted with n-butane where the suppression effect is estimated to be smaller than the precision of the measurement. More information is provided in the Supplement.

The sample preparation and analysis time is 60–65 min. For the in-situ measurements samples are directed onto the first trap by means of a small membrane pump from a continuously-flushed sampling line. In general, each air sample measurement is bracketed by measurements of a quaternary working standard that allows tracking and correction of the MS sensitivity change. The quaternary standards are whole-air samples compressed to 65 bar into 34 L internally electropolished stainless steel canisters (Essex Industries, Missouri, USA). These are collected by the individual groups within AGAGE at various sites during relatively clean air conditions using modified oil-less diving compressors (Rix Industries, USA) or cryogenic techniques. The repeated quaternary standard measurements are used to determine the measurement precisions. For CFC-13 they are ~1.5% (1 $\sigma$) for the Agilent 5973 MSs and ~1% for the newer Agilent 5975 MSs. For CFC-114 the precisions range 0.2% – 0.3% and for CFC-115 0.4% – 0.8%, also showing some improvements with the change to the Agilent 5975 MSs.

As part of the network's calibration scheme and to assess for potential drift of the compounds in the canisters, the quaternary standards are compared once a week on-site against tertiary standards. These are provided by the central calibration facility at SIO and are also whole-air standards in Essex canisters filled under clean air conditions at Trinidad Head or La Jolla (California, USA). These tertiary standards are measured at SIO against secondary whole air standards before they are shipped to the sites and again after their return at the end of their usage times. They are also measured on-site against the previous and next tertiaries. The secondary standards and the synthetic primary standards at SIO provide the core of the AGAGE calibration scheme (Prinn et al., 2000; Miller et al., 2008).



## 2.5 Calibration Scales

AGAGE has been measuring CFC-13 for many years but so far none of these data have been published. This was, among other reasons, due to the use of a primary calibration scale, which was not well defined as it was based on a dilution of a commercial reference gas. The present study prompted the creation of a primary calibration scale for CFC-13 in the ppt range by the Swiss

Federal Institute of Metrology, METAS (Guillevic et al., in preparation). A suite of eleven primary standards was created using a technique that combines permeation tube substance loss determination by a magnetic suspension balance, dynamic dilution through mass flow controllers, and cryogenic collection in containers. These standards covered a range 2.7–4.3 ppt. Comparison between assigned mole fractions and measured relative mole fractions against one of these primary standards revealed an internal consistency of this METAS-2017 calibration scale of 0.6%. AGAGE adopted this calibration scale and all

CFC-13 results reported here are on METAS-2017. It replaced an interim calibration scheme, which was based on a diluted, commercially obtained (Linde) high-concentration standard, and for which a conversion factor of 1.05 was determined.

Measurements of CFC-114 and CFC-115 are reported on the SIO-05 primary calibration scales. They are defined through gravimetric preparations of 13 synthetic primary standards at ambient mole-fraction levels prepared at SIO in 2005 (Prinn et al., 2000). They cover mole fraction ranges of 16–20 ppt for CFC-114 and 8–10 ppt for CFC-115. Internal consistencies for

these sets of standards of 0.14% for CFC-114 and 0.47% for CFC-115 were estimated based on their relative results from intercomparative measurements and their assigned relative mole fractions. Accuracies are initially estimated at 3% ($1\sigma$) for each of the two CFCs and is a conservative estimate based on previous experience with other compounds (a strict statistical treatment of the known uncertainties such as impurities, balance etc would likely lead to a much smaller overall uncertainty). For CFC-114, there is potential for a considerable bias if our results of the combined isomer measurements were to be compared to the

sum of their individual measurements (Supplement). This bias is primarily caused by potentially differing molar sensitivities for the two isomers, the magnitude of which we cannot easily assess on our MSs (see Supplement). Throughout this paper we report all our own measurements as dry air mole fraction (substance fraction) in ppt on these METAS and SIO calibration scales.

In some earlier articles (in particular Sturrock et al. (2001, 2002)), CFC-114 and CFC-115 measurements were published on

calibration scales that were based on diluted, commercially obtained (Linde) high-concentration standards and were referred to as "UB" or "SIO-interim" calibration scales. A later revision resulted in a renaming of these calibration scales to UB-98B for measurements conducted at CSIRO and Cape Grim. After the creation of the SIO-05 primary calibration scales, SIO-05/UB-98B conversion factors of 0.9565 for CFC-114 and 1.0177 for CFC-115 were determined, with which UB-98B-based results need to be multiplied to determine their mole fraction on the SIO-05 calibration scales.

A comparison between the SIO-05 primary calibration scale for CFC-114 and that of the University of East Anglia (UEA) UEA-2014 (Laube et al., 2016) is of limited value and not straight forward because in Medusa-GCMS measurements the two isomers are not chromatographically separated. A detailed discussion on this is given in the Supplement with an intercomparison of the CGAA results measured on both calibration scales. The result of this separate analysis suggests that for near modern (starting about mid-1990s) mole fractions (~16 ppt) numerically summed CFC-114 and CFC-114a mole fractions reported on



the UEA-2014 calibration scales can be converted to SIO-05 reported combined CFC-114 isomer results by multiplication of 1.025.

## 2.6 Uncertainty Assessment for Reported Measurements

To derive accuracies for the reported measurements we combine three independent uncertainties: uncertainties of the calibration
scales mentioned in the previous subsection, a propagation uncertainty, and the instrumental precision of the measured sample, as listed in subsection 2.4. The propagation uncertainties derive from the hierarchical sequence of standards used to propagate assigned mole fraction in the primary standards to the quaternary standards on-site, by assuming measurement uncertainties for each of the steps, i.e. the secondary and tertiary standards. For this step the measurement precisions are assumed the same as those of the quaternary standards on site. For CFC-114 we add an "interference uncertainty", which is based on the findings
of a potentially suppressed MS signal in the presence of n-butane (see Supplement). We estimate a maximum depletion of CFC-114 of 0.6 % in the presence of 0.5 ppb n-butane (which we consider an upper limit in unpolluted air) and add this value as an independent uncertainty. As mentioned earlier (and discussed in the Supplement) there is a potential bias of our combined isomer measurement compared to the sum of individual isomer measurements. However, because differentiation and thus quantification of the bias are not possible for us, it is not included in our uncertainty estimates. The resulting uncertainties
($1\sigma$) for the three compounds are then 3.7% for CFC-13, 3.1% for CFC-114, and 3.2% for CFC-115. They are dominated by the calibration scale uncertainties. For direct comparisons of samples reported on the same calibration scale, the calibration scale uncertainties do not apply and the remaining uncertainties are much smaller (2.2%, 0.7%, and 1.2% for the three compounds, respectively).

## 2.7 Bottom-Up Inventory-Based and Other Emissions Estimates

Here we refer to bottom-up emissions as those derived from data related to production, distribution and usage of these compounds. For the CFCs discussed here such estimates have considerable uncertainties because of the large fraction of these CFCs installed in long-lasting equipment (banks) with unclear leakage rates. Nevertheless bottom-up emission estimates are useful for us as prior for our model analysis and to compare with our top-down observation-based results.

While bottom-up emissions are not available for CFC-13, they were published for CFC-114 and CFC-115 from the refrig-
eration sector by Fisher and Midgley (1993). A more comprehensive set of emissions estimates for these two compounds was released by AFEAS (Alternative Fluorocarbon Environmental Acceptability Study) for 1934–2003. For CFC-114, they show an early onset of emissions in the 1930s with significant released quantities in the late 1940s ($\sim$7 kt yr$^{-1}$) and peak emissions ($\sim$18 kt yr$^{-1}$) in 1986/1987. Extrapolation of the AFEAS data, as in Daniel and Velders (2007), shows emissions of $<$0.1 kt yr$^{-1}$ in the last few years and a remaining bank of 0.16 kt in 2016. On a similar basis, AFEAS CFC-115 bottom-up emissions
started only in the mid 1960s and peaked in the early 1990s at $\sim$13 kt yr$^{-1}$ before declining to $<$0.1 kt yr$^{-1}$ from 2008 leaving a remaining bank of $<$0.01 kt yr$^{-1}$ in 2016. Destruction of these two CFCs is considered insignificant in the AFEAS analysis hence the cumulative production matches the cumulative emissions. Some of these data were used in the Ozone Assessment Report 2006 to produce emission scenarios for 1930–2100 (Daniel and Velders, 2007). Analogously, the AFEAS emission





inventory for CFC-115 was also expanded into a scenario similar to CFC-114, but these results were not graphically presented in the Assessment Report. To facilitate public access to both the AFEAS original numerical data and those expanded in the Assessment Report ("expanded AFEAS data"), we provide these in the Supplement, along with a description of how they were derived. These data are used in the present analysis as priors for the two global inversions.

We also compare our results with the data set derived by Velders and Daniel (2014), who reconstructed production, banks, and emissions for CFC-114 and CFC-115, with projections into the future. Their reconstruction is a mix of bottom-up inventory-based and top-down observation based data. The earlier parts of their records are largely based on the AFEAS results and therefore do not provide significant additional information. Those from 1979–2008 are based on atmospheric observations. Their CFC-114 emissions after 2008 are based on a bank of 15 kt for that year. This bank was derived as a remnant of a 60 kt

bank for 1960, which was 'back'-extrapolated from emissions based on atmospheric observations and using a yearly emission factor (Daniel and Velders (2011), pers. comm. Velders 2017). For CFC-115 Velders and Daniel (2014) derived a bank of 15.9 kt from R-502 for 2008 (UNEP/TEAP (2009) and unpublished data). The Velders and Daniel (2014) bank and emissions after 2008 are significantly larger than those from AFEAS for both compounds.

### 2.8    Firn Model, Global Transport Model, and Inversions

Similar to the study by Vollmer et al. (2016) for halons, the present analysis uses a firn air model to characterize the age of the CFCs in the firn air samples (Trudinger et al., 2016), the AGAGE 12-box model to relate atmospheric mole fractions to surface emissions (Rigby et al., 2013), two inversion approaches to estimate hemispheric emissions, and a Lagrangian transport model to study regional emissions of CFC-115 in north-eastern Asia.

### 2.8.1    Firn Model

The firn model used here was developed at CSIRO by Trudinger et al. (1997) and updated by Trudinger et al. (2013). It has previously been used for firn air measurement reconstructions of other greenhouse gases (Trudinger et al., 2002; Sturrock et al., 2002; Trudinger et al., 2016; Vollmer et al., 2016). Physical processes in the firn, foremost vertical diffusion, cause the air samples to represent age spectra rather than an individual discrete age as is found in e.g. tank samples like the CGAA. Green's functions are used to relate the measured mole fractions to the time-range of the corresponding atmospheric mole fractions.

The update to the firn model described in Trudinger et al. (2013) included a process that had previously been neglected by Trudinger et al. (1997). This process was the upward flow of air due to compression of the pore space as new snow accumulates above, and it appears that this process is important. As discussed in Trudinger et al. (2013), including it in the model removed a discrepancy between DSSW20K firn and CGAA CFC-115 that was noted by Sturrock et al. (2002).

The diffusion coefficients used in this work for the three CFCs relative to $CO_2$ in air (for a temperature of 253 K) are 0.667

for CFC-13 (using le Bas molecular volumes as described by Fuller et al. (1966)), 0.495 for CFC-114 (Matsunaga et al., 1993) and 0.532 for CFC-115 (Matsunaga et al., 1993). Measurement results and reconstructed firn air depth profiles are shown in Fig. 2. These modeled depth profiles are not based on the observations at the individual sites, but rather correspond to the optimized emissions history obtained using measurements from all firn sites as well as the atmospheric measurements used in



this study. While CFC-114 was present in all samples of the three sites, CFC-13 was absent within the detection limits in the South Pole sample and in one of the deepest duplicate samples at DSSW20K. CFC-115 was also absent in two of the three deepest DSSW20K duplicate samples.

### 2.8.2 AGAGE 12-box Model

The AGAGE box model was originally created by Cunnold et al. (1983) and has since been rewritten and modified (Cunnold et al., 1994, 1997; Rigby et al., 2013; Vollmer et al., 2016). In the current version of the model, the atmosphere is divided into four zonal bands, separated at the equator and at the 30° latitudes thereby creating boxes of similar air masses. Boxes are also separated at altitudes represented by 500 hPa and 200 hPa. Model transport parameters, and stratospheric photolytic loss vary seasonally and repeat interannually (Rigby et al., 2013). For the CFCs analyzed here, loss in the atmosphere is dominated by
photolytic destruction in the stratosphere. Here our local stratospheric loss rates are tuned to reflect the current best estimates of the global lifetimes of these compounds from SPARC (2013) as shown in Table 1.

Monthly transport parameters in the 12-box model were tuned to match the simulation of a uniformly distributed passive tracer in the Model for Ozone and Related Tracers (MOZART, Emmons et al. (2010)), using Modern-Era Retrospective Analysis for Research and Application (MERRA) meteorology for the year 2000 (Rienecker et al., 2011). These transport parameters
were repeated each year in our simulations. Whilst inter-annual variation in transport is known to impact the distribution of trace gases in the atmosphere, time-resolved atmospheric physical state estimates are not generally available throughout the entire period of this investigation. Furthermore, we anticipate that variations in emissions dominate atmospheric trends, particularly over the longer (multi-annual) timescales, which are our primary focus.

### 2.8.3 Global Inversions

To estimate global emissions to the atmosphere we employ two different Bayesian inverse methods ("Bristol" and "CSIRO"). Both methods use the AGAGE 12-box model to relate observed tropospheric mole fractions to surface emissions of the CFCs. While past studies using the Bristol inversion have primarily targeted modern in-situ observations from the AGAGE network (Rigby et al., 2011; Vollmer et al., 2011; Rigby et al., 2014; O'Doherty et al., 2014; Vollmer et al., 2015b) the inversion method has also been extended to include firn air observations (Vollmer et al., 2016). Green's functions from the CSIRO firn model are
used in both global inversions to relate the firn air measurements to atmospheric mole fraction over the appropriate time range.

The Bristol approach is based on the methods outlined in Rigby et al. (2011) and extended in Rigby et al. (2014). Briefly, this method assumes a constraint (prior) on the rate of change of emissions, which is adjusted using the data in a Bayesian framework. The magnitude of the uncertainty in the prior year-to-year emissions growth rate is somewhat arbitrarily chosen to be 20% of the maximum emission rate for the entire period. In a minor modification to the approach in Rigby et al. (2014), we
chose to solve for a change in absolute emissions (in kt yr$^{-1}$), rather than a scaling of the prior emissions. This approach was found to lead to more consistent posterior emissions uncertainty estimates between the near-zero and relatively high emissions periods.





The random component of the model-measurement mismatch uncertainties in the Bristol inversion is composed of measurement and calibration scale uncertainties and those of the atmospheric and firn air models. The atmospheric model uncertainty is assumed to be equal to the variability of the estimated baseline within each monthly mean. These uncertainties are propagated through the model to provide a posterior emissions uncertainty estimate (Rigby et al., 2014). The posterior emissions

uncertainty is then augmented with a term related to the calibration scale uncertainty, and the uncertainty due to the lifetime (Rigby et al., 2014). The observations that are compared with the 12-box model in the "Bristol" inversion (see following section) are from all the firn air (firn model output) and CGAA archived air samples described here, and the monthly mean background-filtered in-situ measurements from Mace Head, Trinidad Head, Barbados, American Samoa, and Cape Grim.

The CSIRO inversion, also combined with the 12-box model and Green's functions from the CSIRO firn model (Vollmer et al.,

2016; Trudinger et al., 2016), was developed to focus on sparse observations from air archives, and firn air and ice core samples that are associated with age spectra. The characteristics of these data necessitate the use of regularisation and constraints on the inversion to avoid unrealistic oscillations in the reconstructed mole fractions or negative values of mole fraction or emissions. The CSIRO inversion therefore uses non-negativity constraints and favors relatively small changes to annual emissions in adjacent years rather than large, unrealistic fluctuations. A prior emissions history based on bottom up estimates is used as

a starting point, then a non-linear constrained optimisation method is used to find the solution that minimises a cost function consisting of the model-data mismatch plus the sum of the year-to-year changes in emissions (Trudinger et al., 2016). The observations used in the CSIRO inversion are the firn measurements and annual values of mole fraction from a smoothing spline fit to measurements at Cape Grim and the CGAA, and another spline fit to Mace Head and the NH air archive. Uncertainties are estimated using a bootstrap method that incorporates data uncertainties and uncertainties in the firn model through the use

of an ensemble of firn Green's functions.

We use the expanded AFEAS bottom-up inventory based data for CFC-114 and CFC-115 as prior in the inversions as these were produced without any input from atmospheric observations. For CFC-13, emission inventories do not exist to the best of our knowledge. As prior for this compound, we use the expanded CFC-115 AFEAS data, which we scale with a factor 1/7 based on an intercomparison of production estimates (see Supplement).

**2.8.4 Regional Scale Source Allocation and Atmospheric Inversion**

Pollution events of the three compounds are absent, to within detection limits, from the measurements at all AGAGE field stations with the exception of Gosan (South Korea) and Shangdianzi (China). This has prompted a more detailed analysis of these compounds in north-eastern Asia to locate and quantify potential sources. Only observations from Gosan were used since these are less locally influenced, and therefore less subject to subgridscale model errors, than those from Shangdianzi, which

are subject to pollution events originating in the nearby Beijing capital region.

We calculated qualitative emission distributions by combining model-derived source sensitivities with the above-baseline observations from Gosan. A smooth statistical baseline fit (Ruckstuhl et al., 2012) was subtracted from the observational data. Surface source sensitivities were computed with the Lagrangian Particle Dispersion Model (LPDM) FLEXPART (Stohl et al., 2005) driven by operational analysis/forecasts from the European Centre for Medium-Range Weather Forecasts (ECMWF)


IFS modeling system. 50 000 model particles were released for each 3-hourly time interval and followed backward in time for 10 days. Surface source sensitivities (concentration footprints) were obtained by evaluating the residence times of the model particles along the backward trajectories (Seibert and Frank, 2004).

Qualitative emission distributions were then calculated as a spatially-distributed, weighted concentration average using the source sensitivities as weights. This method is based on the one described by Stohl (1996) for simple air mass trajectories, but was generalized for source sensitivities and applied to halocarbon observations previously (Stemmler et al., 2007; Vollmer et al., 2015a) The method provides a general first impression of potential source locations but cannot be used to quantify individual sources and their uncertainty (location and length).

In addition, we applied a spatially resolved, regional-scale emission inversion, using the same FLEXPART-derived source sensitivities and the Bayesian approach described in detail in Henne et al. (2016). In contrast to the above method, the Bayesian inversion provides a quantitative spatial distribution of posterior emissions and their uncertainties. Prior emissions were set proportional to the population density. The same emission factor per person was used for the entire inversion domain, which comprised most of China, North and South Korea, and the south-western part of Japan. This emission factor was set in such a way that total Chinese emissions were in line with China's share of the gross world product (GWP) of approximately 15% and the global emission estimates described in sections 3.2.1, 3.2.4, and 3.2.6.

Parameters describing the covariance uncertainty matrices were derived from a log-likelihood maximum search (Michalak et al., 2005; Henne et al., 2016). The prior emission uncertainties obtained from this optimization were relatively large and amounted to 0.04, 0.4, and 0.6 kt yr$^{-1}$ for China for CFC-13, CFC-114 and CFC-115, respectively (see Table 2). All analysis was done separately for each year from 2012 to 2016. More details on the applied method and additional results can be found in the Supplement.

## 3 Results and Discussion

### 3.1 Atmospheric Histories and High Resolution Records

We combine our measurement results from firn air samples, archived air in canisters, and in-situ measurements to produce the full historic records for CFC-13, CFC-114, and CFC-115 spanning nearly eight decades (Figs. 3–4). The modeled records discussed in this section derive from the Bristol and CSIRO inversions using these observations and the AGAGE 12-box model. The firn air depth profiles show a steady decline of all three CFCs with increasing depth (Fig. 2). All three compounds are at or below detection limits in the deepest samples of the Antarctic DSSW20K profile but clearly detectable in the deepest samples in the Greenland NEEM-2008 profile. These firn air results are plotted with the full historic record in Fig. 3 using dates based on the effective ages unless the mole fractions were near zero, when mean ages were used; note that these dates are used for graphical purposes only, and that the full Green's functions (shown in the Supplement) were used in the inversions to represent the age of the compounds in firn air. On the temporal scale the firn air results overlap strongly with the results from the archived canisters. These canister samples span from the late 1970s to near-present and overlap with the high-resolution





in-situ measurements shown in more detail in Fig. 4. The in situ data are binned into monthly means after applying a pollution filter to limit the records to samples under background conditions (O'Doherty et al., 2001; Cunnold et al., 2002).

### 3.1.1 CFC-13

CFC-13 first appeared in the atmosphere in the late 1950s to early 1960s (Fig. 3). Between the 1970s and 1980s its growth rates
were highest before declining again in the late 1980s, presumably as a consequence of reduced emissions due to restrictions on production and consumption by the Montreal Protocol in the non-Article 5 countries. Because of its very long lifetime, small emissions are sufficient to maintain the observed increase in its abundance and consequently CFC-13 continued to grow monotonically to a globally averaged mole fraction of 3.18 ppt in 2016. Its global growth rate leveled off at 0.01–0.03 ppt yr$^{-1}$ in the late 1990s. There is, however, no indication of a further decline in the growth rate since then; in contrast, our data suggest
a slight increase during the last decade, which very likely indicates increasing emissions over this period.

Cape Matatula and Cape Grim, which are the two stations most influenced by SH air masses, have shown a small and consistent offset of 0.04 ppt compared to the NH stations (SH lower), which points to predominantly NH emissions (Fig. 4). For the overlapping period of 1978–1997, our CFC-13 abundances are significantly lower (∼25%) compared to earlier published data (Oram, 1999; Culbertson et al., 2004) (Fig. 5).

For most of the AGAGE field stations the high-resolution records show an absence of CFC-13 pollution events indicating vanishings emission within the local footprints of these stations. However Gosan and Shangdianzi feature sporadic pollution events with abundances that reach up to ∼6 ppt. Measurements from the urban sites Tacolneston (England), Dübendorf (Switzerland), and Aspendale (Australia, after removal of a nearby CFC-13 source in early 2010) show no pollution events, while those at La Jolla (USA) exhibit occasional pollution events, which however are smaller and less frequent than those at
the two Asian field sites. High-resolution records are shown in the Supplement.

### 3.1.2 CFC-114

CFC-114 appeared in the atmosphere in the late 1950s to early 1960s at similar times to CFC-13 (Fig. 3). Its growth rate was highest in the 1970s and 1980s (0.5–0.8 ppt yr$^{-1}$) and declined strongly in the 1990s. CFC-114 global mole fractions peaked in 1999–2002 at 16.6 ppt, interhemispheric gradients have vanished since, and global atmospheric mole fractions have slightly
declined since then to 16.3 ppt in 2016. Based on the data assimilated into the Bristol inversion framework, growth rates have been negative since 2000 with a minimum at −0.035 ppt yr$^{-1}$ but increased again since 2010 to −0.010 ppt yr$^{-1}$ by 2016. Similar to CFC-13, this increase likely indicates increased emissions.

Our SH CFC-114 results exhibit lower mole fractions than those of Oram (1999), and those of Sturrock et al. (2002) who used a combination of the data from Oram (1999) and in-situ AGAGE Cape Grim data based on older instrumentation (ADS,
not used in the present study). Also, our CFC-114 abundances are significantly lower than those of Martinerie et al. (2009) for the records before 1980. In contrast, our CGAA mole fractions match the summed CFC-114 isomers mole fractions recently published by Laube et al. (2016) in the older part of the record but get progressively higher (up to 2.5%) for the modern part of the record compared to that reported by Laube et al. (2016).



The high-resolution records at the AGAGE field sites generally show no CFC-114 pollution events indicating vanishing emissions in the airmass footprints of these stations (see Supplement). The single exception is Gosan, which shows frequent pollution events reaching mole fractions of up to ~20 ppt (Shangdianzi data are hampered by an instrumental artifact and not further discussed here). The urban sites also exhibit a similar pattern as discussed for CFC-13 with Tacolneston, Dübendorf

and Aspendale featuring minor and infrequent pollution events indicating that even in these urban areas emissions are small. However the record at La Jolla shows very frequent pollution events with magnitudes up to ~25 ppt. This can be caused by either a rather local source or more widespread emissions in the airmass footprint of this station. A more detailed analysis is beyond the scope of this study.

### 3.1.3  CFC-115

CFC-115 appeared in the atmosphere with a delay of nearly a decade compared to CFC-13 and CFC-114. Its growth peaked at similar times as CFC-13 and the second maximum of CFC-114, at rates of 0.4–0.5 ppt yr$^{-1}$. It then slowed during the mid 2000s to near-zero growth, with CFC-115 abundances leveling off at $\sim 8.3$ ppt in the late 1990s. However, surprisingly, the CFC-115 growth rate has increased since its minimum in 2008 (0.005 ppt yr$^{-1}$) to ~0.02 ppt yr$^{-1}$ in 2015. This has caused an enhanced increase of the atmospheric abundances to 8.49 ppt in 2016, which is seen more clearly in the recent in-situ

measurements from the stations, with increases led by northern hemisphere sites (Fig. 4).

Our CFC-115 abundances agree well with earlier-published results by Culbertson et al. (2004), Oram (1999), Martinerie et al. (2009), and the younger part of the record by Sturrock et al. (2002). The latter study found somewhat larger mole fractions in the 1970s and 1980s, but this was mainly due to a process neglected in the old version of the CSIRO firn model (upward flow of air due to compression, as mentioned above). There are currently no other published data records covering the past two

decades, which we could compare to our results.

The high-resolution records at the AGAGE fields sites show, similar to CFC-13 and CFC-114, no pollution events with the exception of a few rare and small excursions for some of the sites. Again, Gosan and Shangdianzi exhibit pollution events (typically up to 13 ppt), which have become more frequent since  2013 for Gosan, and evident in the Shangdianzi record only starting in 2016 because of missing data for 2013–2016. The urban sites La Jolla and Dubendorf exhibit pollution events,

particularly the former with a more regular frequency. Aspendale showed CFC-115 pollution episodes mainly in 2006–2009 but to a much lesser degree in the most recent part of its record. CFC-115 pollution events are mostly absent from Tacolneston (see Supplement). These observations demonstrate that CFC-115 has not completely been removed from installed equipment in these urban areas.

## 3.2  Emissions

Global emissions of the three CFCs were calculated using the Bristol and CSIRO inversions covering nearly eight decades and are shown in Fig. 6 for 1950 to the present. For CFC-114 and CFC-115, industry-based bottom-up and other reported emissions are available for comparison. For all three CFCs we find persistent lingering emissions in the past decades, which may be expected if release is continuing from banks. However, more unexpected is a recent increase in emissions for all three





substances. In our discussion of these we assume that other effects are absent, which could artificially create these lingering emissions i.e. a slowdown of vertical air mass exchange between the troposphere and the stratosphere and/or reduced removal fluxes. The global emissions results are complemented with results for Asian regional emissions, and with emissions found from specific processes (CFC-13) and compound impurities (CFC-115).

### 3.2.1 CFC-13 Global Emissions

Based on our inversions, CFC-13 emissions increased to a maximum of $\sim$2.6 $\pm$0.25 kt yr$^{-1}$ (1 stdv) in the mid-1980s with a subsequent decline to relatively stable mean emissions of 0.48 $\pm$0.15 kt yr$^{-1}$ during the last decade (Fig 6). Cumulative emissions until 2016 amount to 62 kt (both inversions) of which, due to its 640 yr lifetime (Table 1), $\sim$90% is still in the atmosphere. The persistent emissions over the past two decades are surprisingly high (>10% of the peak emissions). The absence of a clear downward trend over this long period, even more so the recent increase in emissions, are inconsistent with a potential gradual replacement of CFC-13 in refrigeration units after the ban by the Montreal Protocol, which would lead to a decline of CFC-13 banks and emissions. Release functions for the other two CFCs used in the AFEAS vintage model (see Supplement) indicate that emissions of the whole charge of individual refrigeration equipment after installation take 20 years for CFC-114 and 10 years for CFC-115. Assuming similar emission functions for CFC-13 in these applications could potentially explain the decline in the emissions in the 1990s but not the tailing emissions thereafter, about which we can only speculate. One explanation could be different release functions in the last two decades, for example a better containment for some time as a response to reduced availability of CFC-13 for refill, followed by a recent period of enhanced release perhaps due to intensified removal of old refrigeration equipment. Alternatively, CFC-13 could be emitted from sources other than refrigeration systems. It could be a by-product of fluorochemical manufacture and be released from the processes, or as an impurity of the end products. It is unlikely that CFC-13 is used as a process agent as it would need to be recorded and controlled under the regulations of the Montreal Protocol, which is, as far as we know, not the case. The many CFC-13 pollution events measured at Shangdianzi and Gosan, and the rare occurrence at other sites, point to emissions in the East Asian region (although emissions may also be taking place in regions not seen by our high-frequency network).

### 3.2.2 CFC-13 Emissions from Regional FLEXPART Inversion

The transport analysis of CFC-13 pollution peaks that were observed at Gosan did not reveal consistent and localised sources for the years 2012 to 2016. The strongest indication of sources were observed for 2013 and 2014 in China, whereas in 2015 and 2016 no specific source region could be identified (see Supplement). The Bayesian inversion showed relatively weak performance of the simulated prior time series (see Supplement) and the use of the posterior emission field did not improve these simulations to a large extent with the exception of the year 2013 for which a considerable improvement of the simulation was achieved through the emission inversion. Consequently, the posterior emissions of CFC-13 stayed relatively close to the prior estimates with the general tendency of lower posterior estimates for South Korea and Japan (Table 2, Figure 7). Chinese emissions remained very close to the prior estimates except for the year 2013. In summary, these results do not indicate an





over-proportional share of CFC-13 emissions from north-eastern Asia, compared with the global estimate, and, considering the relatively weak model performance, are connected with a considerably large uncertainty.

### 3.2.3  CFC-13 Emissions from Aluminum Smelters

CFC-13 was previously found in the emissions from aluminum plants (Penkett et al., 1981; Harnisch, 1997). Our present CFC-13 study has prompted us to re-analyze emission measurements from an Australian aluminum smelter (Fraser et al., 2013) (see Supplement), and unlike stated in Fraser et al. (2013), we found significant enhancement of CFC-13 in the exhaust samples thereby qualitatively confirming the results from the older studies. Enhancements over background levels of 45 ppt – 130 ppt were found in the various smelter samples. From these results an emission factor of $0.025 \pm 0.017$ g CFC-13 per ton aluminum was calculated. A global extrapolation based on a yearly aluminum production of $\sim$60 Mt yr$^{-1}$ for 2016 (http://www.world-aluminium.org/statistics/#data, accessed June 2017) yields yearly CFC-13 emissions of $0.0015 \pm 0.001$ kt yr$^{-1}$. Unless emission factors from other smelters were significantly higher, this is suggestive of a minor contribution of aluminum smelter emissions to the total global yearly emissions of CFC-13 derived from our atmospheric observations. Nevertheless from a chemical reaction standpoint, these CFC-13 emissions from aluminum production remain unexplained. Simplistically, since the carbon to produce $CF_4$ in aluminium smelters comes from the carbon in the smelter graphite anodes, chlorine impurities in the carbon anodes could be the source of chlorine to produce CFC-13 in the smelters.

### 3.2.4  CFC-114 Global Emissions

Based on our global inversions, CFC-114 emissions started in the 1950s–1960s and reached a maximum in the mid 1970s at $21 \pm 0.28$ kt yr$^{-1}$ followed by a second maximum in 1988 at $22 \pm 0.19$ kt yr$^{-1}$ (Fig. 6). Using the CSIRO inversion we have conducted a sensitivity analysis to investigate the robustness of this double peak. This has shown that the feature remains present when excluding each data set one at a time from the inversion, hence it is not an artifact of the contribution from an individual data set (see Supplement). Global emissions have declined strongly in the 1990s but remained at a surprisingly stable and high level at $1.9 \pm 0.84$ kt yr$^{-1}$ (mean of last decade, see Fig. 6). Our global emissions differ significantly from the AFEAS bottom-up emissions for the first part of the record until about 1980. Bottom-up emissions are significantly elevated compared to our results until 1965 and lower after that. Also, while we derive continuing emissions in the last decade, those from expanded AFEAS data sets were predicted to decline gradually to <0.1 kt yr$^{-1}$ after 2014. Emissions of the last decade reported by Velders and Daniel (2014), which are derived in a bottom-up approach from assumed remaining banks, are higher than those from AFEAS but considerably smaller than the emissions we derived from our observations. Our cumulative emissions until 2016 (587 kt for the Bristol inversion and 586 kt for the CSIRO inversion) are significantly higher than the cumulative emissions and productions derived by AFEAS from an inventory (521 kt) and those reported by Velders and Daniel (2014) (528 kt). Emissions derived by Laube et al. (2016) from atmospheric observations agree well with our emissions after 1980 but with some potential discrepancies in the earlier record (see Laube et al. (2016) for their full emissions record). The cumulative emissions reported by Laube et al. (2016) up to 2014 are 553 kt and agree better with our results than the inventory-based estimates compare with our results.



We can only speculate on these relatively high lingering, and recently increasing emissions of CFC-114. One possible explanation is a change in release functions as was outlined for CFC-13. Alternatively, CFC-114 could be fugitively emitted during synthesis of HFC-134a, where it is an intermediate compound in some synthesis pathways (Rao, 1994; Banks and Sharratt, 1996; McCulloch and Lindley, 2003). We have analyzed a diluted sample of HFC-134a from a container of the high-purity

substance and found CFC-114 present at $2.8 \times 10^{-5}$ mol per mol of HFC-134a. If extrapolated to global HFC-134a emissions of 180 kt yr$^{-1}$ (Rigby et al., 2014) this would correspond to global emissions of 0.084 kt yr$^{-1}$ of CFC-114, which is a minor fraction of the current 1.9 kt yr$^{-1}$. A more comprehensive analysis would be necessary to get an understanding of the variability of such an impurity.

Pollution events in the Asian region, as detected from our high-resolution in-situ measurements, and the absence thereof in

other regions suggest predominant emissions from Asia. However, there is no clear latitudinal gradient in CFC-114 abundance detected from our observations. Nevertheless, increased abundances of CFC-114a (compared to Cape Grim) from samples collected in Taiwan were reported on by Laube et al. (2016), partially supporting our findings.

### 3.2.5   CFC-114 Emissions from Regional FLEXPART Inversion

In contrast to CFC-13, potential emission sources of CFC-114 derived from our observations at Gosan could be identified on the

Chinese mainland for the years after 2013 through the atmospheric transport analysis (see Supplement). The Bayesian regional inversion corroborates this finding, yielding largely increased posterior emissions for China for the years 2014 onwards, with a peak of 1.0±0.2 kt yr$^{-1}$ in 2013 (Table 2, Figure 7). South Korean and Japanese emissions remained around or below the prior values. Posterior emissions were mostly localised in two areas in China, in the Shanghai and its neighboring provinces Zhejiang and Jiangsu, and in the Shandong province (Fig. 8). The overall transport model performance and improvement

through the inversion was largely improved as compared with that for CFC-13 (see Supplement), lending sufficient confidence in the inversion results.

### 3.2.6   CFC-115 Global Emissions

Based on our inversions, CFC-115 emissions started in the mid-1960s and increased to a maximum of 12.8 ±1.3 kt yr$^{-1}$ in the late 1980s. Emissions declined strongly thereafter to a minimum of 0.59 ±0.51 kt yr$^{-1}$ (mean 2007–2010). Surprisingly the

emissions have since increased steadily to 1.14 ±0.50 kt yr$^{-1}$ (mean 2015–2016). Our observations agree very well with the bottom-up emissions by AFEAS except for an earlier maximum in our emissions by a few years compared to that by AFEAS, and for our lingering and increasing emissions over the past years compared to vanishing emissions in the AFEAS record. Consequently, our cumulative emissions until 2016 of 243-245 kt (both inversions) agree well with the cumulative emissions and productions in the expanded AFEAS data of 237 kt and with those by Velders and Daniel (2014) of 228 kt.





### 3.2.7 CFC-115 Emissions from Regional FLEXPART Inversion

The transport analysis of CFC-115 pollution peaks that were observed at Gosan indicated potential emission sources to be mainly located on the Chinese mainland (see Supplement). For the year 2012, no strong sources were located in the domain. Thereafter, potential source locations were identified in the larger Shanghai area (years 2013–2015) and more diffusely from a broad area along the eastern Chinese coast (year 2016). Similarly, the Bayesian inversion yielded increases in posterior emissions mostly located within two areas in Eastern China (Fig. 10), the larger Shanghai area (including the Zhejiang and Jiangsu provinces) and the northern part of the Shandong province. Emissions in these areas showed large posterior emissions for all analyzed years with the most prominent emission hot spots in 2013 and 2014 and smaller posterior emissions in 2012. The locations of increased posterior emissions largely agree with the location of HFC-125 factories (B. Yao, pers. comm. 2017), which we speculate are sources of CFC-115 emissions, see below, and which were not used in the prior. Large posterior emissions in other parts of the domain were not robust, varied from year to year and were also connected with large posterior uncertainties. Total Chinese CFC-115 emissions were estimated to average $0.54\pm0.34\,\mathrm{kt\,yr^{-1}}$ for the years 2013 to 2016 (Table 2, Fig. 7), whereas they remained relatively close to the prior value in 2012 ($0.23\pm0.38\,\mathrm{kt\,yr^{-1}}$). The contribution of those grid cells containing the HFC-125 factories to the total Chinese emissions was considerably increased in the posterior estimates and reached between 12% and 41%, whereas they only contributed 9% in the prior. Posterior estimates for Japan and South Korea did not increase compared to the prior emission estimates.

### 3.2.8 CFC-115 Emissions from HFC-125 Production and Use

We hypothesize that the increased CFC-115 emissions derive, at least in part, from the production of hydrofluorocarbons (HFCs), which have been produced in large quantities during the last two decades. Similar hypotheses were recently put forward for emissions of other CFCs and HCFCs. HCFC-133a (Laube et al., 2014; Vollmer et al., 2015c) and CFC-114a (Laube et al., 2016) emissions were speculated to derive, at least in part, from the production of HFC-134a, and HCFC-31 from the production of HFC-32 (Schoenenberger et al., 2015). The most likely candidate, in the case of CFC-115, is the synthesis of the refrigerant HFC-125. CFC-115 is a known byproduct in one possible pathway to synthesize HFC-125, where tetrachloroethylene is treated with hydrogen fluoride (Rao, 1994; Shanthan Rao et al., 2015) followed by fluorine/chlorine exchanges. Although this pathway is not expected to be widely applied (pers. comm A. McCulloch) there are nevertheless four out of twelve production facilities in China using this route (Chinese Chemical Investment Network, 2017). A possible source is the leakage of CFC-115 to the atmosphere as an intermediate product at factory level, similarly to the speculations put forward in the above studies. However, this hypothesis is difficult to test without factory-level measurements because uncertainties in the localization of "hot spot" emissions, such as the one identified with our regional modeling, exceed the narrow geographical location of a factory potentially present in the area. Nevertheless our analysis of regional emissions agrees with this hypothesis.

In addition to potential CFC-115 emissions at the HFC-125 factory level, we tested the hypothesis of CFC-115 impurities in HFC-125, which would then be emitted to the atmosphere during leakage of HFC-125 in installed refrigeration equipment. In contrast to the cases of HCFC-133a/HFC-134a and HCFC-31/HFC-32, HFC-125 and CFC-115 have similar physico-chemical



properties making it technically difficult to separate the two compounds (Corbin and Reutter, 1997; Kohno and Shibanuma, 2001; Brandstater et al., 2003; Cuzzato and Peron, 2003; Azzali and Basile, 2004; Piepho et al., 2006). We have detected CFC-115 in dilutions of high-purity HFC-125 (Fig. 9). We have also found excess (above ambient) CFC-115 in laboratory air at AGAGE sites at times of air conditioner leakage (Fig. 9). Excess CFC-115 correlated strongly with the main constituents of
the air conditioners (R-410, 50–55% by mass HFC-125, rest HFC-32), but ratios varied depending on site and recharge batch of the air conditioner fluid. These measurements have enabled us to demonstrate impurities ranging from $0.7 - 11 \times 10^{-4}$ mol CFC-115 / mol HFC-125. We extrapolate these to global CFC-115 emissions based on this range and using estimates of global HFC-125 emissions ($40 \, \text{kt yr}^{-1}$, Rigby et al. (2014)) we conclude that this "impurity" source accounts only for $0.0036 - 0.057$ $\text{kt yr}^{-1}$, significantly below the last decade's mean yearly emissions of $0.80 \, \text{kt yr}^{-1}$ derived from our inversions. Note that
HFC-125 in polluted air advected to the sites is generally too low to detect a corresponding CFC-115 enhancement, hence we cannot extend this CFC-115/HFC-125 analysis to the regular air measurements. The CFC-115/HFC-125 ratios in the CFC-115 pollution events observed at Gosan largely exceed the ratios found from air conditioner leakage thereby indicating sources other than HFC-125 impurities.

## 4   Conclusions

Based on a wealth of new observations, we reconstruct the atmospheric histories of CFC-13, CFC-114, and CFC-115 from their first appearance in the atmosphere to 2016. This is the first comprehensive study of the very long lived CFC-13 in the atmosphere. Our global model results suggest that over the last decade the global growth rate for CFC-114 has not declined and those of CFC-13 and CFC-115 have increased, thereby making them two of the few CFCs left with increasing global atmospheric abundances. Under the assumptions of no significant change in global atmospheric transport patterns or sink processes,
these growth rates correspond to ongoing emissions, which have remained stable or even increased over the past decade. This contrasts with the expectations of declining emissions due to the phase-out of these compounds under the regulations of the Montreal Protocol. We provide evidence for small emissions of CFC-114 and CFC-115 as impurities in HFCs and speculate on the possibility of fugitive emissions at the process level. We also find evidence of small emissions of CFC-13 from aluminium smelting, but the chemistry that leads to CFC-13 production is not obvious. Impurities and fugitive emissions are not regulated
by the Montreal Protocol, however even if these are small emissions, they can potentially lead to an increasing atmospheric abundance, particularly for the long-lived CFC-13. For CFC-114 and CFC-115, we find significant emissions from the Asian region but the processes responsible remain largely unknown.

Data used in this study are available from the Supplement, from https://agage.mit.edu/, and from data repositories referenced therein.

*Acknowledgements.*  We acknowledge the station personnel at all stations for their continuous support in conducting in situ measurements, and in flask sampling at Cape Grim (CSIRO and Bureau of Meteorology) and King Sejong (KOPRI). The joint Cape Grim Air Archive (CGAA) project is operated by CSIRO and the Australian Bureau of Meteorology. AGAGE operations at Mace Head, Trinidad Head,





Cape Matatula, Ragged Point, and Cape Grim are supported by the National Aeronautic and Space Administration (NASA) with grants NAG5-12669, NNX07AE89G, NNX11AF17G, and NNX16AC98G to MIT, grants NNX07AE87G, NNX07AF09G, NNX11AF15G, and NNX11AF16G to SIO, through the Department for Business, Energy & Industrial Strategy (BEIS) contract 1028/06/2015 to the University of Bristol for Mace Head and the University of East Anglia for Tacolneston; the National Oceanic and Atmospheric Administration

(NOAA, USA), contract RA-133-R15-CN-0008 to the University of Bristol for Barbados, by the Commonwealth Scientific and Industrial Research Organization (CSIRO Australia), the Bureau of Meteorology (Australia), the Department of Environment and Energy (Australia) and Refrigerant Reclaim Australia. Financial support for the measurements at the other sites is provided; for Jungfraujoch by the Swiss National Program HALCLIM (Swiss Federal Office for the Environment, FOEN) and by the International Foundation High Altitude Research Stations Jungfraujoch and Gornergrat (HFSJG); for Zeppelin by the Norwegian Environment Agency; for Monte Cimone by the National

Research Council of Italy and the Italian Ministry of Education, University and Research through the Project of National Interest Nextdata; for Gosan by the Korea Meteorological Administration Research and Development Program under Grant CATER 2014-6020; and for Shangdianzi by the National Nature Science Foundation of China (41575114). Support for King Sejong flask samples comes from the Swiss State Secretariat for Education and Research and Innovation (SERI) and from the National Resarch Foundation of Korea for the Korean-Swiss Science and Technology Cooperation Program; and from the Korean Polar Research Programs PE13410 and PP16102. CSIRO's contribution

was supported in part by the Australian Climate Change Science Program (ACCSP), an Australian Government Initiative. Australian firn activities in the Antarctic are specifically supported by the Australian Antarctic Science Program. We acknowledge the members of the firn air sampling teams for provision of the samples from Law Dome, NEEM and South Pole. NEEM is directed and organized by the Center of Ice and Climate at the Niels Bohr Institute and US NSF, Office of Polar Programs. It is supported by funding agencies and institutions in Belgium (FNRS-CFB and FWO), Canada (NRCan/GSC), China (CAS), Denmark (FIST), France (IPEV, CNRS/INSU, CEA and ANR),

Germany (AWI), Iceland (RannIs), Japan (NIPR), Korea (KOPRI), The Netherlands (NWO/ALW), Sweden (VR), Switzerland (SNF), United Kingdom (NERC) and the USA (U.S. NSF, Office of Polar Programs). M.K.V. acknowledges a 2011 CSIRO Office of the Chief Executive Distinguished Visiting Scientist grant, and 2016 grants from Empa and SNF to CSIRO Aspendale for technical development and archived air and firn air measurements. M.R. was supported in part by Advanced Research Fellowships from the UK Natural Environment Research Council (NERC, NE/1021365/1).



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





**Table 1.** Evolution of Atmospheric Metrics of CFC-13 (CClF$_3$), CFC-114 (C$_2$Cl$_2$F$_4$), and CFC-115 (C$_2$ClF$_5$).

|  | CFC-13 | CFC-114 | CFC-115 |
|---|---|---|---|
| Ozone Depletion Potential (ODP) |  |  |  |
| ODP, in Montreal Protocol[a] | 1.0 | 1.0 | 0.6 |
| ODP, semiempirical, WMO 2011[b] | – | 0.58 | 0.57 |
| ODP, semiempirical, WMO 2014[c] | – | 0.50 | 0.26 |
| ODP uncertainties, Velders and Daniel (2014)[d] | – | 37%/30% | 34%/32% |
| Global Warming Potential (GWP) [100 yr] |  |  |  |
| GWP, WMO 2011[b] | 14400 | 9180 | 7230 |
| IPCC 2013[e] | 13900 | 8590 | 7670 |
| Velders and Daniel (2014)[f] | – | 9170 (28%) | 6930 (27%) |
| Atmospheric lifetime [yr] |  |  |  |
| Ravishankara et al. (1993)[g] | 640 | 300 | 1700 |
| WMO Ozone Assessment 2006[h] | – | 300 | 1700 |
| WMO Ozone Assessment 2011[i] | – | 190 | 1020 |
| Baasandorj et al. (2013) | – | 214 (210–217) | 574 (528–625) |
| SPARC (2013)[j] | – | 189 (153–247) | 540 (404–813) |
| WMO Ozone Assessment 2014[k] | 640 | 189 (153–247) | 540 (404–813) |

a) Handbook for the Montreal Protocol (UNEP, 2017).

b) WMO Ozone Assessment 2011 (Daniel and Velders, 2011).

c) WMO Ozone Assessment 2014 (Harris and Wuebbles, 2014) using the lifetimes from SPARC (2013) and the fractional release values from Montzka and Reimann (2011).

d) Absolute values as in WMO Ozone Assessment 2014. Uncertainties are $\pm$ for "possible"/"most likely" (on a 95% confidence interval).

e) ICPP (Intergovernmental Panel on Climate Change) 2013 (Myhre et al., 2013) based on Hodnebrog et al. (2013).

f) Updates of WMO Ozone Assessment 2011 with lifetimes from SPARC (2013), and "possible" uncertainty ranges ($\pm$, 95% confidence interval).

g) Ravishankara et al. (1993) give a lower limit value of 380 yr for CFC-13 based on the assumption of a faster vertical mesospheric mixing.

h) WMO Ozone Assessment 2006 (Clerbaux and Cunnold, 2007).

i) WMO Ozone Assessment 2011 (Montzka and Reimann, 2011).

j) Stratosphere-troposphere Processes And their Role in Climate (SPARC), SPARC (2013).

k) WMO Ozone Assessment 2014 (Carpenter and Reimann, 2014).



**Table 2.** By-country emissions of CFC-13, CFC-114, and CFC-115, estimated by the regional inversion: Prior and posterior estimate. All values are given in units of kt yr$^{-1}$; uncertainties represent 1-$\sigma$ range.

| Compound | Year | China | | Hot spots | | South Korea | | Japan | |
|---|---|---|---|---|---|---|---|---|---|
| | | Prior | Posterior | Prior | Posterior | Prior | Posterior | Prior | Posterior |
| CFC-13 | 2012 | 0.1±0.045 | 0.14±0.03 | — | — | 0.004±0.004 | 0.001±0.002 | 0.01±0.008 | 0.007±0.006 |
| CFC-13 | 2013 | 0.1±0.043 | 0.20±0.03 | — | — | 0.004±0.004 | 0.001±0.002 | 0.01±0.008 | 0.007±0.005 |
| CFC-13 | 2014 | 0.1±0.043 | 0.15±0.03 | — | — | 0.004±0.004 | 0.003±0.002 | 0.01±0.008 | 0.007±0.006 |
| CFC-13 | 2015 | 0.1±0.045 | 0.10±0.03 | — | — | 0.004±0.004 | 0.002±0.002 | 0.01±0.008 | 0.004±0.006 |
| CFC-13 | 2016 | 0.1±0.045 | 0.10±0.03 | — | — | 0.004±0.004 | 0.001±0.002 | 0.01±0.008 | 0.004±0.005 |
| CFC-114 | 2012 | 0.4±0.37 | 0.66±0.23 | — | — | 0.015±0.033 | 0.007±0.008 | 0.04±0.065 | 0.019±0.026 |
| CFC-114 | 2013 | 0.4±0.35 | 1.00±0.24 | — | — | 0.015±0.034 | 0.005±0.005 | 0.04±0.068 | 0.033±0.029 |
| CFC-114 | 2014 | 0.4±0.35 | 0.64±0.21 | — | — | 0.015±0.034 | 0.003±0.006 | 0.04±0.067 | 0.017±0.027 |
| CFC-114 | 2015 | 0.4±0.37 | 0.61±0.17 | — | — | 0.015±0.033 | 0.004±0.007 | 0.04±0.065 | 0.034±0.036 |
| CFC-114 | 2016 | 0.4±0.37 | 0.79±0.23 | — | — | 0.015±0.033 | 0.003±0.008 | 0.04±0.065 | 0.019±0.034 |
| CFC-115 | 2012 | 0.2±0.68 | 0.25±0.36 | 0.018±0.13 | 0.046±0.039 | 0.007±0.051 | 0.004±0.008 | 0.02±0.13 | 0.020±0.066 |
| CFC-115 | 2013 | 0.2±0.62 | 0.68±0.25 | 0.018±0.13 | 0.28 ±0.03 | 0.007±0.053 | 0.002±0.005 | 0.02±0.12 | 0.029±0.035 |
| CFC-115 | 2014 | 0.2±0.62 | 0.59±0.26 | 0.016±0.13 | 0.18 ±0.04 | 0.007±0.053 | 0.002±0.006 | 0.02±0.12 | 0.048±0.038 |
| CFC-115 | 2015 | 0.2±0.68 | 0.78±0.35 | 0.018±0.13 | 0.080±0.041 | 0.007±0.051 | 0.002±0.009 | 0.02±0.12 | 0.015±0.032 |
| CFC-115 | 2016 | 0.2±0.62 | 0.47±0.41 | 0.018±0.13 | 0.12 ±0.06 | 0.007±0.053 | 0.005±0.008 | 0.02±0.12 | 0.009±0.029 |





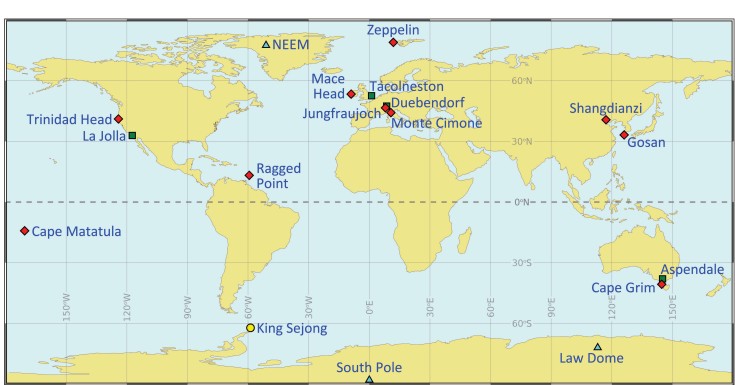

**Figure 1.** Sampling locations for the chlorofluorocarbons CFC-13, CFC-114, and CFC-115 used in this analysis. Filled red diamonds are field sites of AGAGE (Advanced Global Atmospheric Gases Experiment) and related networks, green filled squares are urban sites, the cyan triangles denote the sampling stations for the firn air samples and the yellow filled circle is for the flask sampling site King Sejong, Antarctica.





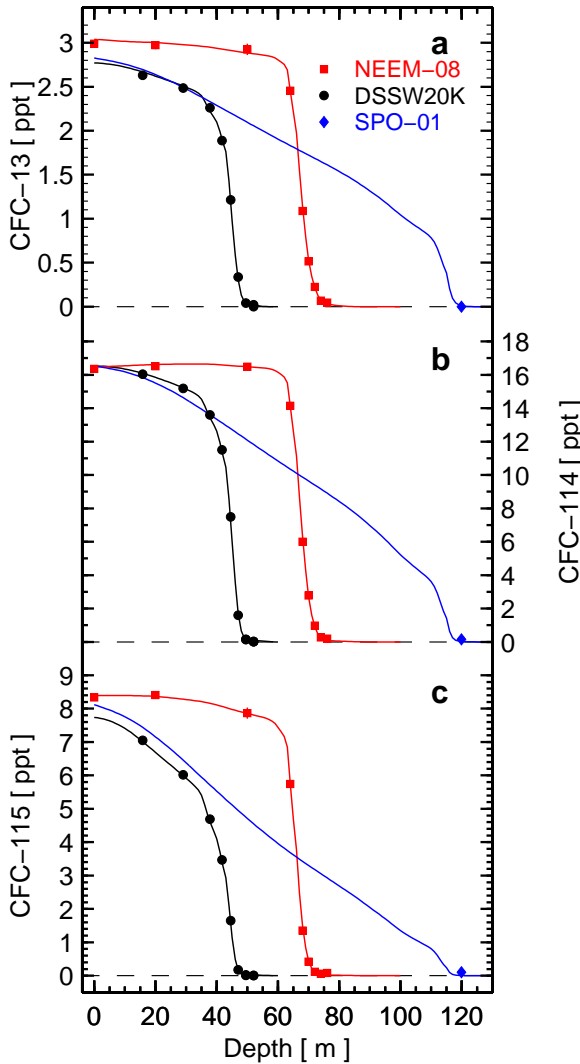

**Figure 2.** Depth profiles for the three chlorofluorocarbons CFC-13 (a), CFC-114 (b), and CFC-115 (c) in polar firn. Measured dry-air mole fractions are shown for the Greenland site NEEM-08 (red squares), and the Antarctic sites Law Dome (DSSW20K, black circles) and South Pole (SPO-01, blue diamond). Generally the measurement precisions ($1\sigma$) are smaller than the plotting symbols. The modeled mole fraction depth profiles (solid lines) correspond to the optimized emissions history from the CSIRO inversion, derived from the combined observations of all three firn sites, archived air and in-situ measurements.





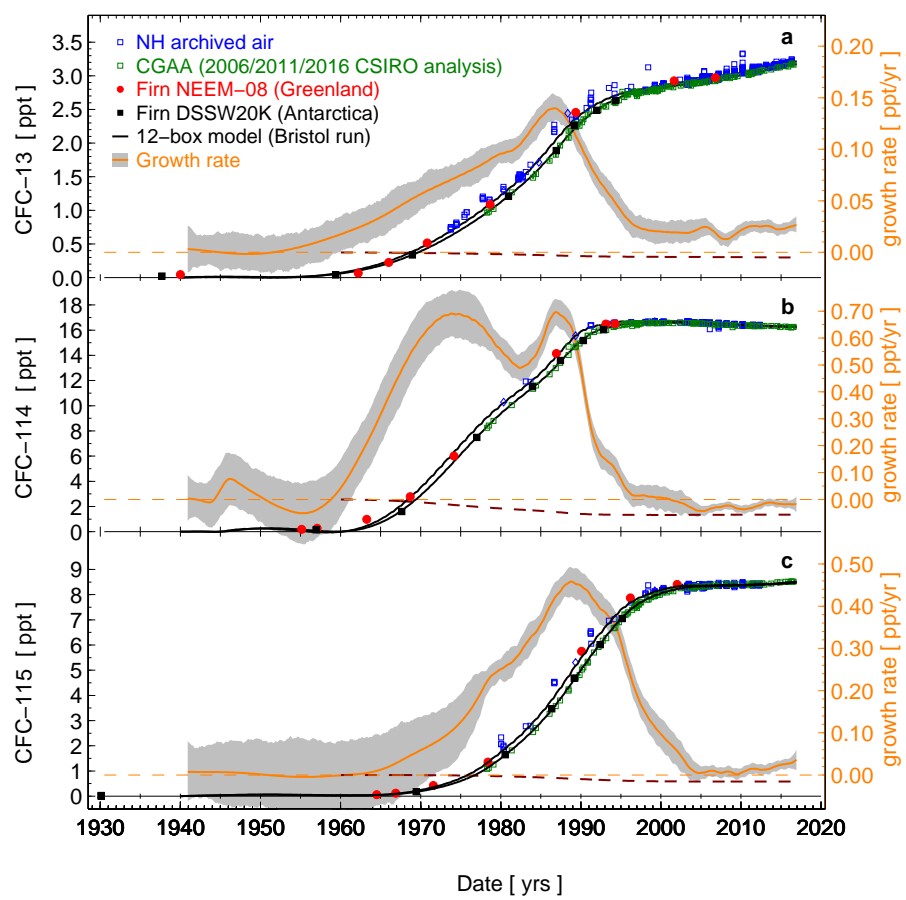

**Figure 3.** Measurements of the chlorofluorocarbons CFC-13 (a), CFC-114 (b), and CFC-115 (c) from archived air samples and firn. Firn measurements are plotted against either effective or mean ages of the samples (see text). In-situ measurement results from the AGAGE stations are not plotted for clarity. The inversion results are given for the northern hemisphere (upper solid lines) and southern hemisphere (lower solid lines). Growth rates (shown in orange using the right axes) are globally averaged from model results. Note that zero growth, shown as dashed orange lines, is offset relative to the left axes. With focus to the recent part of the record these growth rates deviate significantly from the growth rates that would be obtained if zero emissions were assumed (shown as maroon dashed lines, calculated by dividing the global mole fraction by the lifetime).



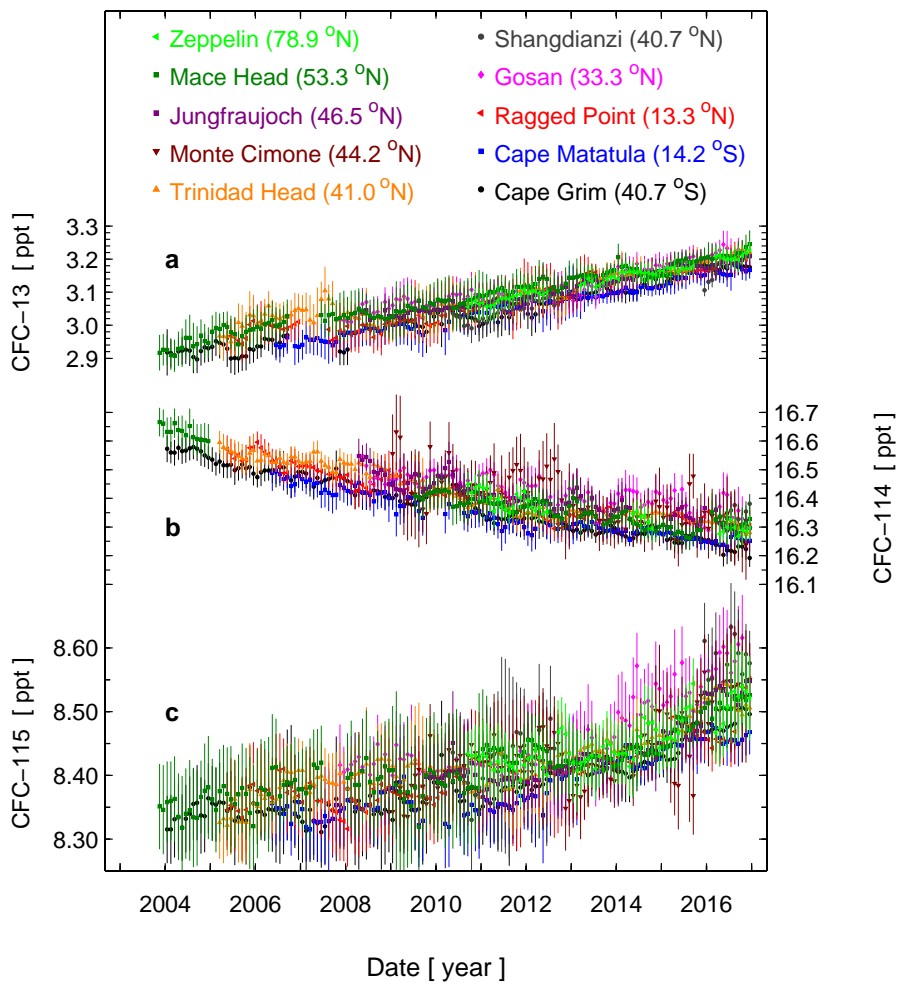

**Figure 4.** Monthly mean abundances of the chlorofluorocarbons CFC-13 (a), CFC-114 (b), and CFC-115 (c) at selected stations of the AGAGE (Advanced Global Atmospheric Gases Experiment) network. Vertical bars are standard deviations of the monthly means (1 $\sigma$) of pollution-filtered observations. Occasional deviations of the Monte Cimone measurements from the other sites for CFC-114 and CFC-115 (CFC-13 not measured at this site) are explained by the significantly larger propagation uncertainties (partially caused by larger precisions) for this site compared to the other site.



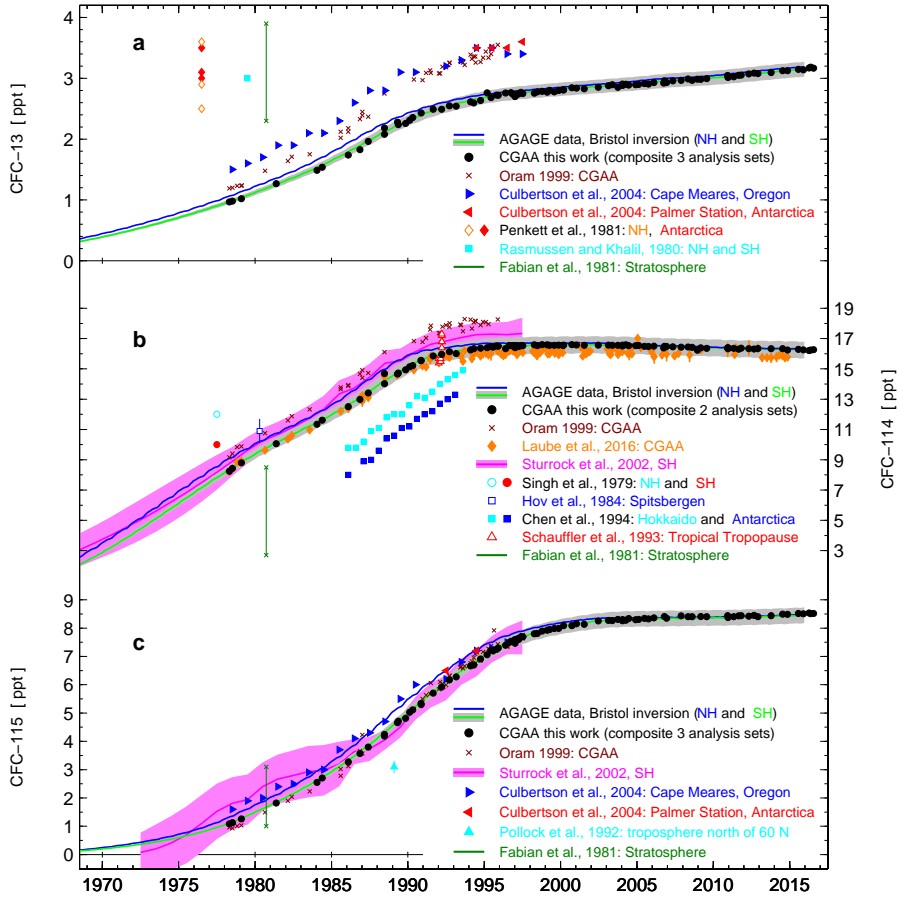

**Figure 5.** Comparison of the atmospheric records of CFC-13 (a), CFC-114 (b), and CFC-115 (c) from this study with previous results. Cape Grim Air Archive (CGAA) samples and subsamples have been analyzed multiple times — here we show analysis results published by Oram (1999), Laube et al. (2016) and the present study (three separate analysis sets are averaged, see Supplement). Light grey bands denote the uncertainty on our SH model results including calibration uncertainty. Uncertainty bands for the NH, which are similar to the SH, are omitted from this plot for clarity. Results for CFC-114 are from combined measurements of the two analytically unseparated CFC-114 isomers. Exceptions to this are the studies by Oram (1999) and Laube et al. (2016) where the numerical sums of the two individual measurements are shown. Also, results by Chen et al. (1994) (values approximated only from their graphical display) are of the $CClF_2CClF_2$ isomer only. Assuming a 5%–6% contribution of CFC-114, these results are still significantly lower compared to our study. All results are left on the calibration scales of the published data.





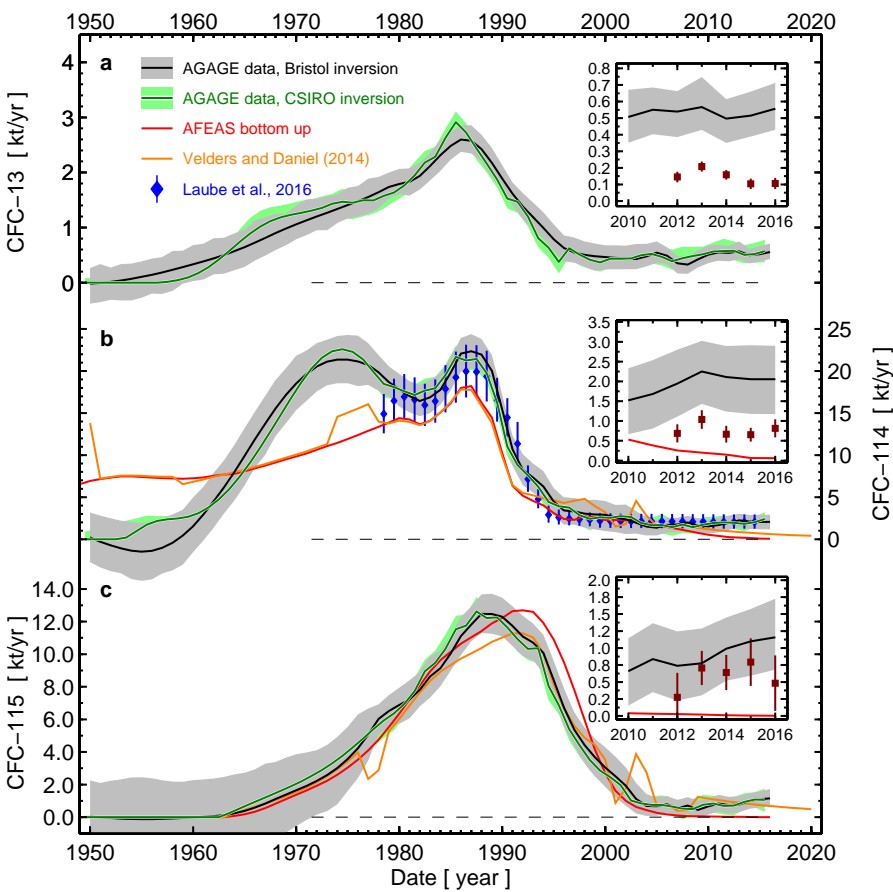

**Figure 6.** Global emissions of the chlorofluorocarbons CFC-13 (a), CFC-114 (both isomers combined) (b), and CFC-115 (c) from atmospheric observations. Black lines and grey shaded areas are for the 'Bristol' inversion and green lines and green shaded areas for the 'CSIRO' inversion (see text). CFC-114 emissions from Laube et al. (2016) are the sum of the emissions of both isomers. In the insets, our observation-based global emissions and the expanded AFEAS bottom-up emissions are compared to the East Asian emissions (maroon diamonds).





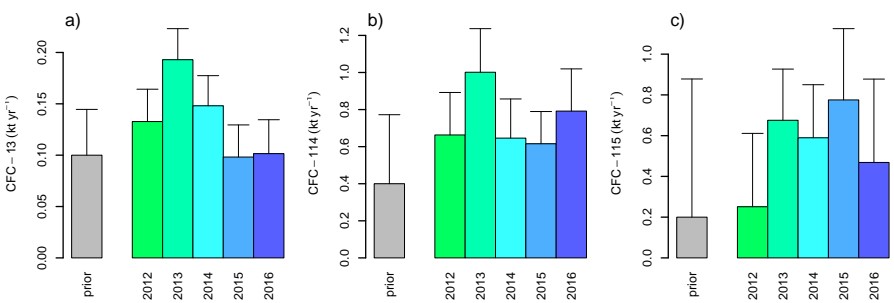

**Figure 7.** Total Chinese emissions of CFC-13, CFC-114, and CFC-115 estimated by the regional inversion. Uncertainties represent 1-$\sigma$ range.

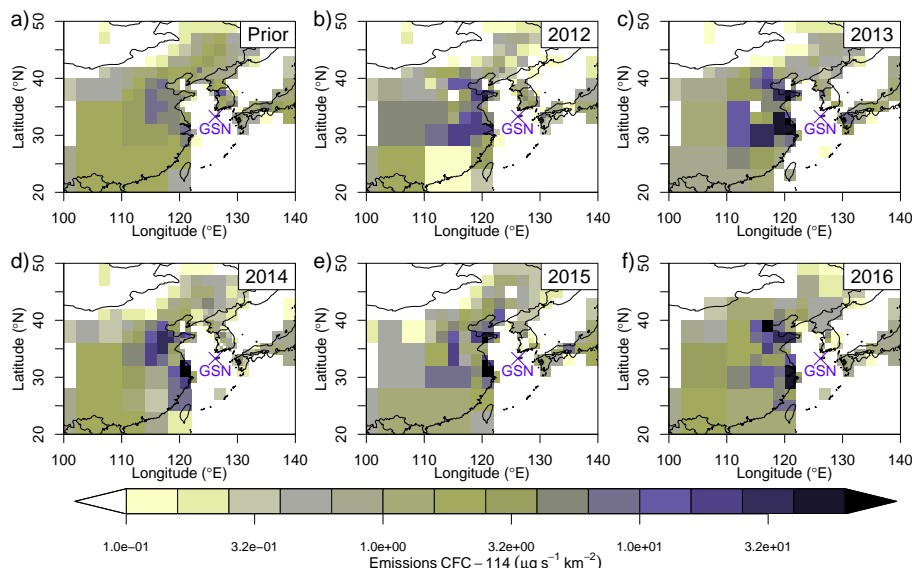

**Figure 8.** Emissions of CFC-114 from north-eastern Asia as estimated by inverse modeling using the CFC-114 observations at Gosan (marked with a blue cross). a) common prior distribution, b-f) posterior distribution for the years 2012 to 2016.





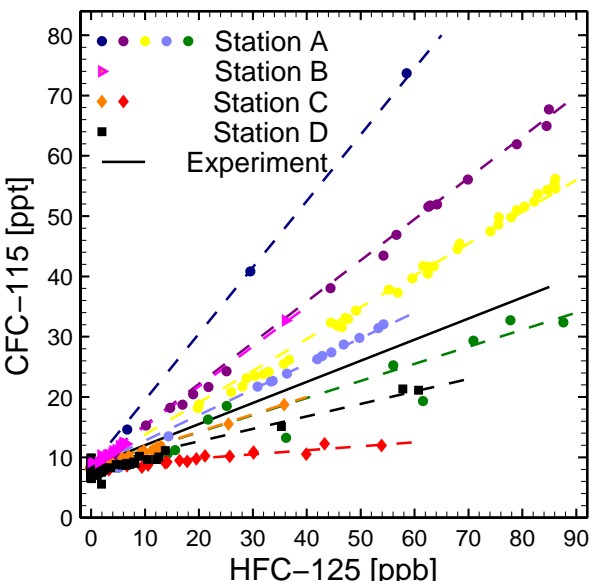

**Figure 9.** CFC-115 contamination in HFC-125 as found in contaminated laboratory air samples at AGAGE stations. Measurements are shown for four stations (A–D) during times of air conditioner leakages (R-410, 50–55% by mass HFC-125, rest is HFC-32). They are plotted for a HFC-125 range $0 – 90$ ppb (parts per billion, $10^{-9}$). Differently-colored episodes are separated by times of air conditioner maintenance and refilling demonstrating the variable fraction of CFC-115 in differing batches of the refrigerant. Based on these observations we find a range from $0.7 – 11 \times 10^{-4}$ mol CFC-115 / mol HFC-125. The solid line ($3.5 \times 10^{-4}$ mol CFC-115 / mol HFC-125) derives from a direct measurement of CFC-115 in a dilution of an independently-obtained sample of pure HFC-125. Dashed lines are approximated only and serve as visual support of the data.





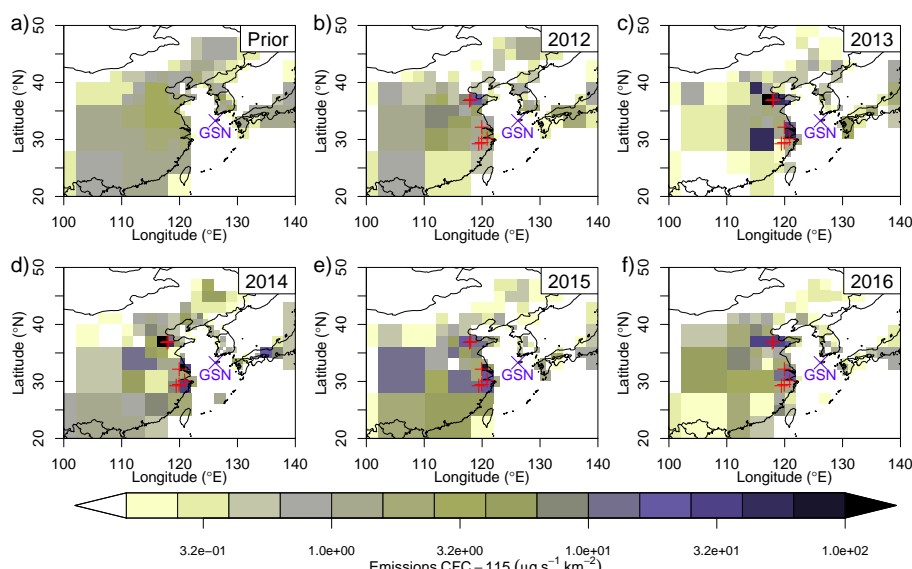

**Figure 10.** Emissions of CFC-115 from north-eastern Asia as estimated by inverse modeling using the CFC-115 observations at Gosan (marked with a blue cross). a) common prior distribution, b-f) posterior distribution for the years 2012 to 2016. Red plus signs mark the location of known HFC-125 factories, which are hypothesized to be potential sources of CFC-115.