# Peer review of "Atmospheric histories and emissions of chlorofluorocarbons CFC-13 (CClF3), $\Sigma$ CFC-114 (C2Cl2F4), and CFC-115 (C2ClF5)"

_Atmospheric Chemistry and Physics, 2017_

## Referee Comment (RC1) · Anonymous Referee #1 · 10 Nov 2017

This paper describes the atmospheric histories of CFC-13, CFC-114, and CFC-115; substances controlled under the Montreal Protocol on Substances that Deplete the Ozone Layer. The authors present atmospheric measurements and measurements of air archived in cylinders and firn, and use these to estimate historical emissions. They also investigate regional emissions using high-frequency measurements at at site in Korea.

The results are significant in that they represent the first comprehensive study of atmospheric CFC-13. The results also complement a recent studies of CFC-114. The Vollmer et al study includes new information on possible sources these gases, including emission as impurities in other gases used in refrigeration. This study provides constraints on current emissions of gases controlled under the Montreal Protocol, and recent (possibly unexpected) increases in emissions.

General Comments

This paper is comprehensive, well-written, and based on well-established methods. I do not have any objections to publication. The overall body of work and technical information available in the Supplement will be of interest to others in this field.

Specific Comments

Table 1: Why not include GWP from WMO 2014 (CFC-114, CFC-115)? Also, the lifetime of CFC-13 was reported as 640 yr in WMO 2006 and WMO 2011, but is not listed in Table 1.

Table 2: How do you define a "hot spot"? It looks like there could be "hot spot" emissions of CFC-114 also in 2013.

Pg. 8, Line 2: You use the term "primary calibration scale". Consider using "interim calibration scale" instead since you refer to the "interim" scale on line 10.

Pg 9, Line 28: Add "(See Supplement)" after "Extrapolation of the AFEAS data, as in Daniel and Velders (2007)"

Pg. 9, Line 33: I don't see emission scenarios from 1930-2100 in the 2006 Assessment Report. Do you mean atmospheric abundances from 1990-2040 (Fig 8-2) or 1955-2100 (Table 8-5)?

Pg 12. Line 11: What is meant by "regularization"? Do you simply mean "additional constraints"?

Pg 13, Line 14: Can you comment on the sensitivity of posterior emissions to the magnitude of the priors?

Pg. 15, Line 33: "more unexpected" than what?

Pg. 17, Line 29: Suggest "projected by Velders and Daniel ...." since 2016 emissions would have been a projection in 2014

Pg. 18, Line 10: Do you mean that a "change" in the latitudinal gradient has not been detected? There is clearly a gradient (N-S).

Pg 18, Line 19: Possibly re-phase. The use of "improvement/improved" in same sentence not entirely clear. Or refer to Supplement for model performance?

Figure 2: I'm not sure how the blue line (SPO) adds to the story, being based on only one sample from the SPO firn. Since this paper does not focus heavily on firn results, it might be better to keep the SPO sample "point", but delete the "line".

Figure 6: Hard to tell the difference between orange and red lines.

Figure S4: panels "b" and "c" look very similar. Perhaps draw a circle around green points in "c" to draw attention to what is different?

Pg S18, Line 11: Something still missing [Cathy to calculate this value]?

Pg. 17, Line 4: Seems like a sentence is needed here to clarify that CFC-13 emissions were not reported by Fraser et al, 2013 (if that is what you are saying). I suggest: "CFC-13 was previously found in the emissions from aluminum plants (Penkett et al., 1981; Harnisch, 1997), but was not reported by Fraser et al (2013) from a similar study." And then follow with: "On re-analysis of the Fraser et al samples, we found enhancements over background levels of 45 ppt – 130 ppt in the various smelter samples. "

Technical Corrections

Pg 7, Line 17: Add comma between "measurements" and "samples"

Pg 14, Line 5: delete "again"

Pg 15, line 11: Suggest: "Its growth rate then slowed during the mid-2000s to near

zero, with . . ."

Pg 16, Line 3: Suggest substitute "removal rates" for "removal fluxes"0

Pg. 18, Line 25: add "since 2010" after "steadily"

Pg. 19, Line 7: Suggest: "Large posterior emissions were detected for all analyzed years . . .."

Pg. 19, Line 13: I calculate a different number (0.63 kt) for the average Chinese emissions in the years 2013-2016 from values in Table 2 (0.68, 0.59, 0.78, 0.47).

Pg. 19, Line 13: Total Chinese emissions in 2012: (0.23±0.38 kt yr1) does not match value shown in Table 2 for 2012.

Fig. 9 caption: change "derives from" to "was derived from"

Fig. S4: "Laube et al 2017" should be "Laube et al 2016" (two places in figure, caption is correct)

Pg. S16, Line 8: Suggest replacing "emissions are rather faster" with "emissions occur earlier in the life-cycle"

Pg. S17, Line 12: This sentence does not read well. "The very crude approach we have taken is still based on the above assumption of similarities to CFC-115 but are comparing production data, . . .". Do you mean " . . .. but is based on a comparison of production data"?

---

## Referee Comment (RC2) · Anonymous Referee #2 · 15 Nov 2017

This is an impressive, comprehensive, thought-provoking, and very useful paper that describes the full atmospheric histories over eight decades for three CFCs that are present in the atmosphere at much lower concentrations than the primary CFCs. It documents continued emissions and, surprisingly, notable increases in global emissions of CFC-115 that are rightfully indicated as being difficult to explain given the global production phase-out for CFCs. New sources for emissions of these gases are identify and, using inverse analyses of high-frequency atmosphere data, continuing emissions of CFC-114 and -115 from East Asia (China) are inferred in amounts that account for a large portion of the ongoing global emissions.

The only substantive issue I have with the manuscript relates to two conclusions that I'm not convinced are defensible, given our uncertain knowledge of lifetimes. Once these are reconsidered, I think the paper is ready to publish.

Those issues: 1) on cumulative emission vs production comparisons (abstract and text), it seems necessary to express the influence of uncertain lifetimes on the differences argued for here for CFC-114. I pretty sure the lifetime uncertainty and hence uncertainty on derived global emissions is much larger than 10%, implying that one can't reliably conclude that there is a 10% discrepancy here or, by implication, evidence for significant unreported production. The authors might also consider if a discussion of bank magnitudes instead of cumulative emissions provides more insight (discussed below). 2) the assertion that emissions of CFC-114 and CFC-13 have increased in recent years. It is not at all clear from the figures of mole fraction rate of change or derived global emissions that the authors are correct in stating that emissions of these gases have actually increased in recent years beyond the variability and uncertainty envelope of recent years.

While that abstract states that CFC-115 impurity in HFC-125 production cannot account for all of the ongoing CFC-115 emission, consider also stating that it isn't likely that this impurity source, given an average impurity content of10e-3 to 10e-4 in HFC-125, is the cause of the identified CFC-115 emission increase in recent years (at least this is what I conclude looking at the numbers).

Consider mentioning 500-yr GWPs, given that this is the timeframe for atmospheric destruction for these gases so is certainly relevant... Along those lines, consider mentioning how CFC-115 impurities affect the 500-yr GWP of HFC-125 use (small effect it seems, on average).

Discussion of bank sizes relative to the current emission rate as a fraction of peak emissions. CFC-12 recently has been ∼10%, CFC-11 is higher... It mostly comes down to the size of banks and release rates from those banks.

The authors have done a good job of discussing the issue of CFC-114 and CFC-114a being measured as one chemical in this work and in most previous studies. The text is still confusing in places, however, as CFC-114 is used to indicate the sum of both and just the one isomer in studies in which separation was accomplished (in caption of Figure 5, to name one spot; in discussion of lifetime too, are lifetimes stated for 114 or is it 114+114a here? To clarify this I'd suggest not using CFC-114 to mean CFC-114 + CFC-114a; consider CFC-114* or CFC-114s (where s=sym as in Supplement) or something else to refer to the sum. I presume the presence of 114a with 114 relates to unavoidable co-production during synthesis and an inability to separate these isomers before sales, so that the presence of both in perhaps changing source ratios relates to different production pathways. I didn't see this mentioned, or missed it if it was.

Reconsider the discussion of banks and "cumulative emissions", as these seem different ways to express the outcome of essentially similar analyses. Yet it is difficult to ascertain the differences from the previous analyses (e.g., p. 9, line 28-31, bank determinations for 2016 are indicated to be v. small for 114 and 115 based on AFEAS emission histories and an analysis by Daniel and Velders (2007), that presumably considered atmospheric measurements in some way) and what the authors find here, which is expressed as "cumulative emissions" rather than an implied bank magnitude. This is relevant for 114, in particular, given the quite different history derived here compared to what has been considered in the past. I'm guessing that the new results suggest a negative bank for CFC-114 recently and a minimal bank for CFC-115 (i.e., <1 year's worth of emission). Typically, having an estimate of bank size is useful to consider for understanding current emission rates, and availing yourself of this would seem a useful addition. In the case of 114 and 115 it seems clear that any emission from banks are less likely to be the source of these ongoing emissions, since bank sizes estimated here are negligible (albeit their magnitude is not precisely known owing to lifetime uncertainty).

What I find very curious is the peak emission derived for CFC-114 the 1970s (the first

peak). This peak may generate the apparent negative bank today (assuming lifetimes are accurate). While a discussion is included to suggest it is a robust result, no discussion of its plausibility is presented. Given that this history is very different than the production-derived emission history, if it is accurate does it suggest emissions from a process unrelated to reported production (byproduct emission)? Does the timing correspond to other chlorinated & fluorinated ethane production histories, such as CFC-113? HCFC-142b had unusually high concentrations in the early CCAA record—is the timing of that similar or not? Is it possible that the CFC114&114a sum is causing trouble here (the Supplement indicates that the error created could be an offset in time), given that the true ratio is not known before 1975?

Figure 3. I don't understand why the "zero growth" line is included in the figure. This line has little meaning other than to indicate steady-state, which isn't emphasized in paper, and it's clearly evident which side of steady state emissions are on currently, given one look at Figure 4. It seems to me the important reference line to retain here is the one indicating growth for zero emission–this other line I find distracting.

Figure 4 caption: the term "pollution filtered" isn't entirely clear. I presume you mean background atmospheric concentrations? Consider a different term.

p. 2, line 2. Consider additions and changes: "larger than would be expected from zero emissions \*\*given currently estimated lifetimes\*\*". Also, "unaltered" is ambiguous meaning here. I think you mean "constant" or "unchanging".

p. 3, line 35. This isn't true apriori. Be sure to comment on the lifetime difference for 114a and 114 to make clear to the reader if it is an important factor in affecting changes in the relative atmospheric abundance of these gases. p. 17, line 30, was the same lifetime used in Laube et al? Seems important to consider before discussing potential differences.

p S18, line 11 (Supplement text) don't forget to add the missing value here.

Regarding firn results, I didn't see the detection limit for instrumentation mentioned in the text or supplement, but would be useful to add.

---

## Author Comment (AC1) · 6 Dec 2017

Reply to : Anonymous Referee #1

This paper describes the atmospheric histories of CFC-13, CFC-114, and CFC-115; substances controlled under the Montreal Protocol on Substances that Deplete the Ozone Layer. The authors present atmospheric measurements and measurements of air archived in cylinders and firn, and use these to estimate historical emissions. They also investigate regional emissions using high-frequency measurements at at site in Korea.

The results are significant in that they represent the first comprehensive study of atmospheric CFC-13. The results also complement a recent studies of CFC-114. The Vollmer et al study includes new information on possible sources these gases, including emission as impurities in other gases used in refrigeration. This study provides constraints on current emissions of gases controlled under the Montreal Protocol, and recent (possibly unexpected) increases in emissions.

> Reply: We thank the referee for his/her thorough comments and have provided some answers below, which we have incorporated into a revised manuscript that would be ready to distribute in case the editor decides to proceed to the next review step with this manuscript.

General Comments
This paper is comprehensive, well-written, and based on well-established methods. I do not have any objections to publication. The overall body of work and technical information available in the Supplement will be of interest to others in this field.

Specific Comments
Table 1: Why not include GWP from WMO 2014 (CFC-114, CFC-115)? Also, the lifetime of CFC-13 was reported as 640 yr in WMO 2006 and WMO 2011, but is not listed in Table 1.

> Reply: These suggestions are now added to the table.

Table 2: How do you define a "hot spot"? It looks like there could be "hot spot" emissions of CFC-114 also in 2013.

> Reply: We have used the term "hot spot" in a loosely way of describing an area with enhanced emissions. We have now removed the wording in one place in the text and described it a bit more in detail in Table 2. It is true that there are also areas of enhanced emissions for CFC-114 but these are by far not as pronounced as for CFC-115, and for CFC-114 we do not have corresponding factory locations at hands (as the HFC-125 factory locations we have for CFC-115).

Pg. 8, Line 2: You use the term "primary calibration scale". Consider using "interim calibration scale" instead since you refer to the "interim" scale on line 10.

> Reply: done, this makes it more clear.

Pg 9, Line 28: Add "(See Supplement)" after "Extrapolation of the AFEAS data, as in Daniel and Velders (2007)"

> Reply: Done

Pg. 9, Line 33: I don't see emission scenarios from 1930-2100 in the 2006 Assessment Report. Do you mean atmospheric abundances from 1990-2040 (Fig 8-2) or 1955-2100 (Table 8-5)?

> Reply: Emission scenarios were used to derive atmospheric abundances. To clarify this, we change the text to "*Some of these data were used in the Ozone Assessment Report 2006 to*

*produce emission scenarios for 1930--2100 on which the atmospheric abundances for the same period were based (Daniels and Velders, 2007)*".

Pg 12. Line 11: What is meant by "regularization"? Do you simply mean "additional constraints"?

Reply: We have now removed that part and it now reads: *"The characteristics of these data necessitate the use of constraints on the inversion to avoid unrealistic oscillation in the reconstructed mole fractions or negative values of mole fraction or emissions."* In the Supplement we have described the regularization for the coefficient alpha in more detail.

Pg 13, Line 14: Can you comment on the sensitivity of posterior emissions to the magnitude of the priors?

Reply: We analyzed the sensitivity of our inversion results towards the magnitude of our prior. For differences of the prior of +/- 50 % we don't get a strong response in the a-posteriori emissions. For some years the a-posteriori emissions remain practically independent of the prior, whereas for other years the differences resulting from different priors are well within the uncertainty estimates of the a-posteriori (see figure below). We added the following sentence to the revised manuscript (in the Supplement):

"*The choice of the magnitude of the prior emission was tested by running additional sensitivity inversions with 50% higher and lower prior emissions. The influence on the a-posteriori emissions was small compared to the a-posteriori uncertainty estimate.*"

[Figure]

Pg. 15, Line 33: "more unexpected" than what?

Reply: This is indeed poorly phrased: We have now rephrased this part (also following some suggestions of another reviewer), and the text now reads: *"For all three CFCs we find persistent lingering emissions in the past decades. While the emissions for CFC-13 and CFC-114 have remained stable within uncertainties, those for CFC-115 have increased in recent years."*

Pg. 17, Line 29: Suggest "projected by Velders and Daniel . . .." since 2016 emissions would have been a projection in 2014

Reply: Done

Pg. 18, Line 10: Do you mean that a "change" in the latitudinal gradient has not been detected? There is clearly a gradient (N-S).

Reply: This is indeed a wrong statement we have made. We suggest to change these sentences to: *"The observed latitudinal gradient in CFC-114 abundance suggests predominant NH emissions. Pollution events in the Asian region, as detected from our high-resolution in-situ measurements, and the absence thereof in other regions suggest that at least some of these emissions originate from Asia. Increased abundances of CFC-114a, compared to Cape Grim,*

*from samples collected in Taiwan were reported on by Laube et al. (2016), partially supporting our findings."*

Pg 18, Line 19: Possibly re-phase. The use of "improvement/improved" in same sentence not entirely clear. Or refer to Supplement for model performance?

Reply: We agree and have revised the sentence. It now reads: *"The overall transport model performance and its improvement through the inversion (see Supplement) were considerably better as in the case of CFC-13, lending sufficient confidence in the inversion results."*

Figure 2: I'm not sure how the blue line (SPO) adds to the story, being based on only one sample from the SPO firn. Since this paper does not focus heavily on firn results, it might be better to keep the SPO sample "point", but delete the "line".

Reply: We prefer to keep the line in there, leaving it out would make the reader wonder why it is not there. It also helps to put the SPO point into perspective.

Figure 6: Hard to tell the difference between orange and red lines.

Reply: Agreed, we have now changed the colors and hope that the two lines can now be distinguished more easily.

Figure S4: panels "b" and "c" look very similar. Perhaps draw a circle around green points in "c" to draw attention to what is different?

Reply: We agree, but we have decided to highlight the difference in a different way by changing colors and symbol sizes. We have now also modified the caption to clarify the purpose of this figure.

Pg S18, Line 11: Something still missing [Cathy to calculate this value]?

Reply: Thank you for spotting this, this is now fixed, it reads: "*The difference it makes to the CSIRO inversion, when we consider only zero mole fraction at 52 m DSSW20K, or with both the zero and non-zero value, is within the uncertainties in emissions.*"

Pg. 17, Line 4: Seems like a sentence is needed here to clarify that CFC-13 emissions were not reported by Fraser et al, 2013 (if that is what you are saying). I suggest: "CFC-13 was previously found in the emissions from aluminum plants (Penkett et al., 1981; Harnisch, 1997), but was not reported by Fraser et al (2013) from a similar study." And then follow with: "On re-analysis of the Fraser et al samples, we found enhancements over background levels of 45 ppt – 130 ppt in the various smelter samples. "

Reply: In the Fraser et al. (2013) publication, CFC-13 emissions were reported as absent, which we discovered to be an erroneous statement. We have now rephrased this part, hoping that it is now clear that Fraser et al. (2013) reported zero CFC-13 emissions but that our re-analysis showed significant CFC-13 emissions.

Technical Corrections

Pg 7, Line 17: Add comma between "measurements" and "samples"

Reply: done

Pg 14, Line 5: delete "again"

Reply: done

Pg 15, line 11: Suggest: "Its growth rate then slowed during the mid-2000s to near zero, with . . ."

Reply: Changed according to the suggestion, but left the word "rate" out.

Pg 16, Line 3: Suggest substitute "removal rates" for "removal fluxes"

Reply: We prefer to keep the term "fluxes" as the term "rates" could here be confusing and e.g. mistaken for chemical reaction rates.

Pg. 18, Line 25: add "since 2010" after "steadily"

Reply: done

Pg. 19, Line 7: Suggest: "Large posterior emissions were detected for all analyzed years . . .."

Reply: We agree and changed the sentence accordingly.

Pg. 19, Line 13: I calculate a different number (0.63 kt) for the average Chinese emissions in the years 2013-2016 from values in Table 2 (0.68, 0.59, 0.78, 0.47).

Reply: This is an embarrassing mistake we made, it is now corrected --- thanks for spotting it.

Pg. 19, Line 13: Total Chinese emissions in 2012: (0.23±0.38 kt yr1) does not match value shown in Table 2 for 2012.

Reply: Same as above, and we have now cross-checked other numbers as well to ensure that not more error of that kind are present.

Fig. 9 caption: change "derives from" to "was derived from"

Reply: We changed that phrase to: "… is the result from ..."

Fig. S4: "Laube et al 2017" should be "Laube et al 2016" (two places in figure, caption is correct)

Reply: Thank you for spotting this, it is now fixed.

Pg. S16, Line 8: Suggest replacing "emissions are rather faster" with "emissions occur earlier in the life-cycle"

Reply: We agree and have now changed to the suggested wording.

Pg. S17, Line 12: This sentence does not read well. "The very crude approach we have taken is still based on the above assumption of similarities to CFC-115 but are comparing production data, . . .". Do you mean " . . .. but is based on a comparison of production data"?

Reply: Agreed. We changed the wording of several sentences in this paragraph to provide more clarity.

---

## Author Comment (AC2) · 6 Dec 2017

Reply to Anonymous Referee #2

This is an impressive, comprehensive, thought-provoking, and very useful paper that describes the full atmospheric histories over eight decades for three CFCs that are present in the atmosphere at much lower concentrations than the primary CFCs. It documents continued emissions and, surprisingly, notable increases in global emissions of CFC-115 that are rightfully indicated as being difficult to explain given the global production phase-out for CFCs. New sources for emissions of these gases are identify and, using inverse analyses of high-frequency atmosphere data, continuing emissions of CFC-114 and -115 from East Asia (China) are inferred in amounts that account for a large portion of the ongoing global emissions.

> Reply: We thank the referee for his/her thorough comments and have provided some answers below, which we have incorporated into a revised manuscript that would be ready to distribute in case the editor decides to proceed to the next review step with this manuscript.

The only substantive issue I have with the manuscript relates to two conclusions that I'm not convinced are defensible, given our uncertain knowledge of lifetimes. Once these are reconsidered, I think the paper is ready to publish.

Those issues:
1) on cumulative emission vs production comparisons (abstract and text), it seems necessary to express the influence of uncertain lifetimes on the differences argued for here for CFC-114. I pretty sure the lifetime uncertainty and hence uncertainty on derived global emissions is much larger than 10%, implying that one can't reliably conclude that there is a 10% discrepancy here or, by implication, evidence for significant unreported production. The authors might also consider if a discussion of bank magnitudes instead of cumulative emissions provides more insight (discussed below).

> Reply: We agree with the reviewer that the uncertainties in the lifetimes are far too big to conclude on a significant discrepancy between cumulative emissions and production. We have now deleted the corresponding sentence from the abstract and will limit the discussion on this to the text. Then in the text, we change the wording. For CFC-114, we remove the word 'significantly' and add a comment, such that it now reads: "*Our cumulative emissions until 2016 (587 kt for the Bristol inversion and 586 kt for the CSIRO inversion) are higher than the cumulative emissions and productions derived by AFEAS from an inventory (521 kt) and those projected by Velders and Daniel (2014) (528 kt). However, despite this ~10% difference they agree within the large uncertainties, in particular those of the CFC-114 lifetime (Table 1).*" With regard to comparison of the cumulative emissions to those by Laube et al. (2016), see further below, where the reviewer has addressed this specifically. As for CFC-115, we write that our cumulative emissions agree with production within the large uncertainties of the lifetime.

2) the assertion that emissions of CFC-114 and CFC-13 have increased in recent years. It is not at all clear from the figures of mole fraction rate of change or derived global emissions that the authors are correct in stating that emissions of these gases have actually increased in recent years beyond the variability and uncertainty envelope of recent years.

> Reply: We agree with the reviewer. We have previously assumed that the uncertainty envelope is dominated by lifetime uncertainties and primary calibration uncertainties, which both would need to be excluded for the assessment of a potential trend in the growth rates and emissions. However when excluding these systematic uncertainties in an additional analysis, stimulated by the reviewer's comment, the overall uncertainties do not get significantly smaller. We therefore

agree with the reviewer and removed all statements related to increase of growth rates and/or emissions for CFC-13 and CFC-114 in recent years. We now state that they have not declined (within the uncertainties).

While that abstract states that CFC-115 impurity in HFC-125 production cannot account for all of the ongoing CFC-115 emission, consider also stating that it isn't likely that this impurity source, given an average impurity content of 10e-3 to 10e-4 in HFC-125, is the cause of the identified CFC-115 emission increase in recent years (at least this is what I conclude looking at the numbers).

Reply: This is correct and we decided to include a statement about this in the abstract and in the text. This required some re-arranging of sentences, and it reads now: "*We find impurities of CFC-115 in the refrigerant HFC-125 (CHF2CF3) but if extrapolated to global emissions, they can neither account for the lingering global CFC-115 emissions determined from the atmospheric observations, nor for their recent increases.*"

Consider mentioning 500-yr GWPs, given that this is the timeframe for atmospheric destruction for these gases so is certainly relevant... Along those lines, consider mentioning how CFC-115 impurities affect the 500-yr GWP of HFC-125 use (small effect it seems, on average).

Reply: This is a good point. We have now added the 500-yr GWPs to the Table 1 (Climate Metrics) and mention the large 500-yr GWP for CFC-13 in the introduction.

As for the second part (effect on 500-yr GWP of HFC-125), this is an interesting thought. However since this would be about the topic HFC-125, it would not add to the CFC stories, and given that this manuscript is already rather long, we decide to not elaborate on this topic. We agree that the effect would probably be small. It also raises the question on what compounds/life cycle effects should be added to the "process" HFC-125 use – if impurities should be added, then should e.g. emissions (e.g. CO2) of other parts of the life cycle (installation, maintenance) be added to that "HFC-125 use".

Discussion of bank sizes relative to the current emission rate as a fraction of peak emissions. CFC-12 recently has been ~10%, CFC-11 is higher... It mostly comes down to the size of banks and release rates from those banks.

Reply: Yes, we agree on this. Nevertheless, it is surprising that for CFC-13, it is >10% of the peak emission, or >15% if only the last few years are considered. And remaining high. Given the ban and assuming that the life-cycle of CFC-13 is similar to CFC-114 or CFC-115, this is not really expected. We do agree with the reviewer, a few years ago, CFC-12 has also been around 10% of the peak emissions, which is somewhat surprising also. CFC-11 cannot really be compared with these compounds because of its installment in foam – its life-cycle is much different, which is likely causing a less pronounced peak and a longer tail, which is still a large fraction (20%) of the peak. We don't see a need to change the text with regard to this comment.

The authors have done a good job of discussing the issue of CFC-114 and CFC-114a being measured as one chemical in this work and in most previous studies. The text is still confusing in places, however, as CFC-114 is used to indicate the sum of both and just the one isomer in studies in which separation was accomplished (in caption of Figure 5, to name one spot; in discussion of lifetime too, are lifetimes stated for 114 or is it 114+114a here? To clarify this I'd suggest not using CFC-114 to mean CFC-114 + CFC-114a; consider CFC-114* or CFC-114s (where s=sym as in Supplement) or something else to refer to the sum. I presume the presence of 114a with 114 relates to unavoidable co-production during synthesis and an inability to separate these isomers before sales, so that the presence of both in perhaps changing source ratios relates to different production pathways. I didn't see this mentioned, or missed it if it was.

Reply: We agree that some of the terminology is still confusing. We have therefore adopted a new nomenclature for this article after intense discussions among co-authors, and we now refer to ΣCFC-114 as the combined (mainly Medusa GCMS) measurements of CFC-114 and CFC-114a. This comment by the reviewer, and an additional information, which became available on

the CFC-114/CFC-114a of the reference material used to produce the primary standards, have led us to completely re-write the Supplement S.3-4. The reviewer's comment on whether the lifetime statements apply to CFC-114 or CFC-114 + CFC-114a is very stimulating. In fact this question applies to the climate metrics (e.g. Table 1) in general. We have now investigated this further by consulting original literature of laboratory experiments, from which some of these climate metrics have been deduced. Interestingly even though these studies clearly target CFC-114 (CClF2CClF2) alone, there is no mentioning of potential CFC-114a impurities in their reference material, or any comment related to potential purification. Nevertheless it is well know that it is very difficult to produce pure CFC-114, and there are examples of impurities found at significant levels (5%). Such an impurity would have had an impact on the UV absorption spectra and ultimately the lifetime and other climate metrics. We have now added some comments on this in the introduction, and we extended Table 1 by stating that the values we cite from the literature should be for CFC-114, but could be biased.

Reconsider the discussion of banks and "cumulative emissions", as these seem different ways to express the outcome of essentially similar analyses. Yet it is difficult to ascertain the differences from the previous analyses (e.g., p. 9, line 28-31, bank determinations for 2016 are indicated to be v. small for 114 and 115 based on AFEAS emission histories and an analysis by Daniel and Velders (2007), that presumably considered atmospheric measurements in some way) and what the authors find here, which is expressed as "cumulative emissions" rather than an implied bank magnitude. This is relevant for 114, in particular, given the quite different history derived here compared to what has been considered in the past. I'm guessing that the new results suggest a negative bank for CFC-114 recently and a minimal bank for CFC-115 (i.e., <1 year's worth of emission). Typically, having an estimate of bank size is useful to consider for understanding current emission rates, and availing yourself of this would seem a useful addition. In the case of 114 and 115 it seems clear that any emission from banks are less likely to be the source of these ongoing emissions, since bank sizes estimated here are negligible (albeit their magnitude is not precisely known owing to lifetime uncertainty).

Reply: As the reviewer stated earlier, lifetime uncertainties are large and an assessment of implied bank size by comparing with the bottom-up emissions/banks is tagged with these uncertainties. From our descriptions of the banks in Section 2.7 it becomes obvious that those for AFEAS are negligibly small and those by Daniel and Velders are also not in line with the currently high emissions that we observe. As the reviewer states, the bank consideration is essentially a different way of expressing what we already state. We cannot state that the results suggest a negative bank for CFC-114 given the large uncertainties in the lifetime. The comparison of the cumulative emissions up to 2016 is, again within the lifetime uncertainties, a valuable approach to interpret the very different CFC-114 emission histories we find compared to the bottom-up approach. Nevertheless we add a sentence for each CFC-114 and CFC-115, where we compare our recent emissions with the banks of AFEAS and Velders and Daniel for 2016, to re-iterate in slightly different words, that the recent emissions we estimate are not in line with the model of bank and emissions by these bottom up approaches. The suggestion of the reviewer that "any emission from banks are less likely to be the source of these ongoing emissions" is a statement that we cannot support as such. We do not know if some shorter term banks have been built up in recent years, and hence we would need to launch a discussion on long-term and short-term banks, which does not seem appropriate here. For ΣCFC-114 the additional sentence now reads: "*Our estimates of the annual emissions for the recent years are large compared to the banks proposed by AFEAS (0.2 kt for 2016) and Velders and Daniel (2014) (6.4 kt for 2016) and are suggestive of additional recently-produced ΣCFC-114.*" Similarly, the additional sentence we propose for CFC-115 is: "Nevertheless, our annual emissions for the recent years are large compared to the banks proposed by AFEAS (<0.01 kt for 2016) and Velders and Daniel (2014) (8.6 kt for 2016) and are suggestive or additional, recently-produced CFC-115.

What I find very curious is the peak emission derived for CFC-114 the 1970s (the first peak). This peak may generate the apparent negative bank today (assuming lifetimes are accurate). While a discussion is included to suggest it is a robust result, no discussion of its plausibility is presented. Given that this history is very different than the production-derived emission history, if it is accurate

does it suggest emissions from a process unrelated to reported production (byproduct emission)? Does the timing correspond to other chlorinated & fluorinated ethane production histories, such as CFC-113? HCFC-142b had unusually high concentrations in the early CCAA record. Is the timing of that similar or not? Is it possible that the CFC114&114a sum is causing trouble here (the Supplement indicates that the error created could be an offset in time), given that the true ratio is not known before 1975?

Reply: The referee is correct in saying that there is no plausibility discussion on the occurrence of two emission peaks over time. We would find any potential explanation as too speculative and hence prefer to not elaborate on these in the paper. It is not a-priori unrealistic that emissions had changed in that way from the main use of ΣCFC-114, perhaps there was better containment for a while, or excessive emissions (causing the first peak) or temporal changes in the productions and uses. While the idea of byproduct emissions at that time is interesting, we find it too speculative to be mentioned here. It is not possible that the CFC-114a/CFC-114 is causing trouble here, the potential errors involved with this are too small and are also gradual. Unlike the reviewer states, the true CFC-114a/CFC-114 ratio is somewhat known for before 1975, see firn air results by Laube et al., 2016 (also compare measured isomers in the firn (supplement) and modeled ratio in their main paper).

Figure 3. I don't understand why the "zero growth" line is included in the figure. This line has little meaning other than to indicate steady-state, which isn't emphasized in paper, and it's clearly evident which side of steady state emissions are on currently, given one look at Figure 4. It seems to me the important reference line to retain here is the one indicating growth for zero emission–this other line I find distracting.

Reply: The "zero growth" line serves both the purposes of providing a visual aid of where zero is in these plots (similar to the horizontal black lines for providing a zero line for the abundances), and to denote the steady-state balance between emissions and destructions. It is correct that some of this can be seen in Fig. 4, but we find it important to have this information in the same figure as the actual growth rates. Also Fig. 4 goes only back to 2004, so for example the "crossing over" for CFC-114 in about 1996 is not seen in Fig. 4. Further, the slow-down in the decline of CFC-114 abundances (or the increase in the growth rate) can be much better viewed in the presence of that "zero growth" reference line. Also for example for CFC-115, the "zero growth" line for the most recent years is outside the full uncertainty bands for the last few years, which is another information one would not get from Fig. 4. For these reasons we have decided to leave these "zero growth" lines in the figure, but we now change the plotting scheme (colors, lines) to avoid confusion and distraction, and we have modified the legend.

Figure 4 caption: the term "pollution filtered" isn't entirely clear. I presume you mean background atmospheric concentrations? Consider a different term.

Reply: We agree that this has not been termed clearly and we have now changed the text to read: "*Abundances of the chlorofluorocarbons CFC-13 (a), ΣCFC-114 (b), and CFC-115 (c) at selected stations of the AGAGE (Advanced Global Atmospheric Gases Experiment) network. These in situ data are binned into monthly means after applying a pollution filter to limit the records to samples under background conditions (O'Doherty et al., 2001; Cunnold et al., 2002). Vertical bars are standard deviations of the monthly means (1 sigma).*"

p. 2, line 2. Consider additions and changes: "larger than would be expected from zero emissions **given currently estimated lifetimes**". Also, "unaltered" is ambiguous meaning here. I think you mean "constant" or "unchanging".

Reply: We believe that the "given currently estimated lifetimes" is implicit. We have changed "unaltered" to "unchanging".

p. 3, line 35. This isn't true a-priori. Be sure to comment on the lifetime difference for 114a and 114 to make clear to the reader if it is an important factor in affecting changes in the relative atmospheric abundance of these gases.

Reply: We agree that a comment on the lifetimes needs to be included to make this statement correct. We suggest a rephrasing to: "*These results showed for the first time an increasing ratio of CFC-114a/CFC-114 in the atmosphere, and, given a shorter lifetime of CFC-114a compared to CFC-114, pointing to an increasing CFC-114a/CFC-114 emission ratio over time*".

p. 17, line 30, was the same lifetime used in Laube et al? Seems important to consider before discussing potential differences.

Reply: This is a good point. Regardless of what Laube et al 2016 have used for the (symmetric) CFC-114 lifetime (looks like they have used 189 yrs, same as we have used), they have used compound-dependent lifetimes (i.e. for CFC-114a they have used 102 yrs), while we have used the 189 yrs for the combined $\Sigma$CFC-114 measurements. Also, primary calibration scales are different for the two networks, which we deliberately do not correct for in our Fig. 6. Given these considerations, a quantitative comparison (as we had done in our first submission) of our (cumulative) emissions to those by Laube is questionable and hence we decided to delete that statement. Leaving it in there would have caused a lot of hand-waving and would have made the manuscript longer again.

p S18, line 11 (Supplement text) don't forget to add the missing value here.

Reply: done, it now reads "*The difference it makes to the CSIRO inversion, when we consider only zero mole fraction at 52 m DSSW20K, or with both the zero and non-zero values, is within the uncertainties in emissions*.

Regarding firn results, I didn't see the detection limit for instrumentation mentioned in the text or supplement, but would be useful to add.

Reply: Detection limits are given in the supplement Table S1. We now refer to that in the text by extending the sentence on flag values to "Further description of the flag values and the detection limits are given in Table S1. ".